# Black-Box Certification with Randomized Smoothing: A Functional Optimization Based Framework

**Dinghuai Zhang**[*]
Mila
dinghuai.zhang@mila.quebec

**Mao Ye**[*], **Chengyue Gong**[*]
Department of Computer Science
University of Texas at Austin
{my21, cygong}@cs.utexas.edu

**Zhanxing Zhu**
School of Mathematical Sciences
Peking University
zhanxing.zhu@pku.edu.cn

**Qiang Liu**
Department of Computer Science
University of Texas at Austin
lqiang@cs.utexas.edu

## Abstract

Randomized classifiers have been shown to provide a promising approach for achieving certified robustness against adversarial attacks in deep learning. However, most existing methods only leverage Gaussian smoothing noise and only work for $\ell_2$ perturbation. We propose a general framework of adversarial certification with non-Gaussian noise and for more general types of attacks, from a unified functional optimization perspective. Our new framework allows us to identify a key trade-off between accuracy and robustness via designing smoothing distributions and leverage it to design new families of non-Gaussian smoothing distributions that work more efficiently for different $\ell_p$ settings, including $\ell_1$, $\ell_2$ and $\ell_\infty$ attacks. Our proposed methods achieve better certification results than previous works and provide a new perspective on randomized smoothing certification.

## 1 Introduction

Although many robust training algorithms have been developed to overcome adversarial attacks [1, 2, 3], most heuristically developed methods can be shown to be broken by more powerful adversaries eventually (*e.g.,* [4, 5, 6, 7]). This casts an urgent demand for developing robust classifiers with provable worst-case guarantees. One promising approach for certifiable robustness is the recent *randomized smoothing method* [8, 9, 10, 11, 12, 13, 14, 15], which constructs smoothed classifiers with certifiable robustness by introducing noise on the inputs. Compared with the other more traditional certification approaches [16, 17, 18] that exploit special structures of the neural networks (such as the properties of ReLU), the randomized smoothing approaches work more flexibly on general black-box classifiers and is shown to be more scalable and provide tighter bounds on challenging datasets such as ImageNet [19].

Most existing methods use Gaussian noise for smoothing. Although appearing to be a natural choice, one of our key observations is that the Gaussian distribution is, in fact a sub-optimal choice in high dimensional spaces even for $\ell_2$ attack. We observe that there is a counter-intuitive phenomenon in high dimensional spaces [20], that almost all of the probability mass of standard Gaussian distribution concentrates around the sphere surface of a certain radius. This makes tuning the variance of Gaussian distribution an inefficient way to trade off robustness and accuracy for randomized smoothing.

---

[*]Equal contributions

**Our Contributions**   We propose a general framework of adversarial certification using non-Gaussian smoothing noises, based on a new functional optimization perspective. Our framework unifies the methods of [9] and [14] as special cases, and is applicable to more general smoothing distributions and more types of attacks beyond $\ell_2$-norm setting. Leveraging our insight, we develop a new family of distributions for better certification results on $\ell_1$, $\ell_2$ and $\ell_\infty$ attacks. An efficient computational approach is developed to enable our method in practice. Empirical results show that our new framework and smoothing distributions outperform existing approaches for $\ell_1$, $\ell_2$ and $\ell_\infty$ attacking, on datasets such as CIFAR-10 and ImageNet.

## 2   Related Works

**Certified Defenses**   Unlike the empirical defense methods, once a classifier can guarantee a consistent prediction for input within a local region, it is called a certified-robustness classifier. *Exact* certification methods provide the minimal perturbation condition which leads to a different classification result. This line of work focuses on deep neural networks with ReLU-like activation that makes the classifier a piece-wise linear function. This enables researchers to introduce satisfiability modulo theories [21, 22] or mix integer linear programming [23, 24]. *Sufficient* certification methods take a conservative way and bound the Lipschitz constant or other information of the network [18, 16, 25, 26]. However, these certification strategies share a drawback that they are not feasible on large-scale scenarios, *e.g.* large and deep networks and datasets.

**Randomized Smoothing**   To mitigate this limitation of previous certifiable defenses, improving network robustness via randomness has been recently discussed [27, 28]. [8] first introduced randomization with technique in differential privacy. [12] improved their work with a bound given by Rényi divergence. In succession, [9] firstly provided a *tight* bound for *arbitrary* Gaussian smoothed classifiers based on previous theorems found by [29]. [10] combined the empirical and certification robustness, by applying adversarial training on randomized smoothed classifiers to achieve a higher certified accuracy. [11] focused on $\ell_0$ norm perturbation setting, and proposed a discrete smoothing distribution which can be shown perform better than the widely used Gaussian distribution. [14] took a similar statistical testing approach with [9], utilizing Laplacian smoothing to tackle $\ell_1$ certification problem. [15] extended the approach of [9] to a top-k setting. [13] extends the total variant used by [9] to $f$-divergences. Recent works [30, 31, 32] discuss further problems about certification methods. We also focus on a generalization of randomized smoothing, but with a different view on loosing the constraint on classifier.

Noticeably, [30] also develops analysis on $\ell_1$ setting and provide a thorough theoretical analysis on many kinds of randomized distribution. We believe the [30] and ours have different contributions and were developed concurrently. [30] derives the optimal shapes of level sets for $\ell_p$ attacks based on the Wulff Crystal theory, while our work, based on our functional-optimization framework and accuracy-robustness decomposition (Eq.9), proposes to use distribution that is more concentrated toward the center. Besides, we also consider a novel distribution using mixed $\ell_2$ and $\ell_\infty$ norm for $\ell_\infty$ adversary, which hasn't been studied before and improve the empirical results.

## 3   Black-box Certification as Functional Optimization

### 3.1   Background

**Adversarial Certification**   For simplicity, we consider binary classification of predicting binary labels $y \in \{0, 1\}$ given feature vectors $x \in \mathbb{R}^d$. The extension to multi-class cases is straightforward, and is discussed in Appendix C. We assume $f^\sharp \colon \mathbb{R}^d \to [0, 1]$ is a given binary classifier ($\sharp$ means the classifier is *given*), which maps from the input space $\mathbb{R}^d$ to either the positive class probability in interval $[0, 1]$ or binary labels in $\{0, 1\}$. In the robustness certification problem, a testing data point $\boldsymbol{x}_0 \in \mathbb{R}^d$ is given, and one is asked to verify if the classifier outputs the same prediction when the input $\boldsymbol{x}_0$ is perturbed arbitrarily in $\mathcal{B}$, a given neighborhood of $\boldsymbol{x}_0$. Specifically, let $\mathcal{B}$ be a set of possible perturbation vectors, *e.g.*, $\mathcal{B} = \left\{ \boldsymbol{\delta} \in \mathbb{R}^d : \|\boldsymbol{\delta}\|_p \leq r \right\}$ for $\ell_p$ norm with a radius $r$. If the classifier predicts $y = 1$ on $\boldsymbol{x}_0$, i.e. $f^\sharp(\boldsymbol{x}_0) > 1/2$, we want to verify if $f^\sharp(\boldsymbol{x}_0 + \boldsymbol{\delta}) > 1/2$ still holds for any $\delta \in \mathcal{B}$. Through this paper, we consider the most common adversarial settings: $\ell_1$, $\ell_2$ and $\ell_\infty$ attacks.

**Black-box Randomized Smoothing Certification**  Directly certifying $f^\sharp$ heavily relies on the smooth property of $f^\sharp$, which has been explored in a series of prior works [16, 8]. These methods typically depend on the special structure-property (*e.g.*, the use of ReLU units) of $f^\sharp$, and thus can not serve as general-purpose algorithms for any type of networks. Instead, We are interested in *black-box* verification methods that could work for *arbitrary* classifiers. One approach to enable this, as explored in recent works [9, 11], is to replace $f^\sharp$ with a smoothed classifier by convolving it with Gaussian noise, and verify the *smoothed* classifier.

Specifically, assume $\pi_0$ is a smoothing distribution with zero mean and bounded variance, e.g., $\pi_0 = \mathcal{N}(\mathbf{0}, \sigma^2)$. The randomized smoothed classifier is defined by

$$f^\sharp_{\pi_0}(\boldsymbol{x}_0) := \mathbb{E}_{\boldsymbol{z} \sim \pi_0}\left[f^\sharp(\boldsymbol{x}_0 + \boldsymbol{z})\right],$$

which returns the averaged probability of $\boldsymbol{x}_0 + \boldsymbol{z}$ under the perturbation of $\boldsymbol{z} \sim \pi_0$. Assume we replace the original classifier with $f^\sharp_{\pi_0}$, then the goal becomes certifying $f^\sharp_{\pi_0}$ using its inherent smoothness. Specifically, if $f^\sharp_{\pi_0}(\boldsymbol{x}_0) > 1/2$, we want to certify that $f^\sharp_{\pi_0}(\boldsymbol{x}_0 + \boldsymbol{\delta}) > 1/2$ for every $\boldsymbol{\delta} \in \mathcal{B}$, that is, we want to certify that

$$\min_{\boldsymbol{\delta} \in \mathcal{B}} f^\sharp_{\pi_0}(\boldsymbol{x}_0 + \boldsymbol{\delta}) = \min_{\boldsymbol{\delta} \in \mathcal{B}} \mathbb{E}_{\boldsymbol{z} \sim \pi_0}[f^\sharp(\boldsymbol{x}_0 + \boldsymbol{z} + \boldsymbol{\delta})] > \frac{1}{2}. \tag{1}$$

In this case, it is sufficient to obtain a *guaranteed lower bound* of $\min_{\boldsymbol{\delta} \in \mathcal{B}} f^\sharp_{\pi_0}(\boldsymbol{x}_0 + \boldsymbol{\delta})$ and check if it is larger than $1/2$. When $\pi_0$ is Gaussian $\mathcal{N}(\mathbf{0},\ \sigma^2)$ and for $\ell_2$ attack, this problem was studied in [9], which shows that a lower bound of

$$\min_{\boldsymbol{z} \in \mathcal{B}} \mathbb{E}_{\boldsymbol{z} \sim \pi_0}[f^\sharp(\boldsymbol{x}_0 + \boldsymbol{z})] \geq \Phi(\Phi^{-1}(f^\sharp_{\pi_0}(\boldsymbol{x}_0)) - \frac{r}{\sigma}), \tag{2}$$

where $\Phi(\cdot)$ is the cumulative density function (CDF) of standard Gaussian distribution. The proof of this result in [9] uses Neyman-Pearson lemma [29]. In the following section, we will show that this bound is a special case of the proposed functional optimization framework for robustness certification.

## 3.2 Constrained Adversarial Certification

We propose a **constrained adversarial certification (CAC)** framework, which yields a guaranteed lower bound for Eq.1. The main idea is simple: assume $\mathcal{F}$ is a function class which is known to include $f^\sharp$, then the following optimization immediately yields a guaranteed lower bound

$$\min_{\boldsymbol{\delta} \in \mathcal{B}} f^\sharp_{\pi_0}(\boldsymbol{x}_0 + \boldsymbol{\delta}) \geq \min_{f \in \mathcal{F}} \min_{\boldsymbol{\delta} \in \mathcal{B}} \left\{ f_{\pi_0}(\boldsymbol{x}_0 + \boldsymbol{\delta}) \ \text{s.t.} \ f_{\pi_0}(\boldsymbol{x}_0) = f^\sharp_{\pi_0}(\boldsymbol{x}_0) \right\}, \tag{3}$$

where we define $f_{\pi_0}(\boldsymbol{x}_0) = \mathbb{E}_{\boldsymbol{z} \sim \pi_0}[f(\boldsymbol{x}_0 + \boldsymbol{z})]$ for any given $f$. Then we need to search for the minimum value of $f_{\pi_0}(\boldsymbol{x}_0 + \boldsymbol{\delta})$ for all classifiers in $\mathcal{F}$ that satisfies $f_{\pi_0}(\boldsymbol{x}_0) = f^\sharp_{\pi_0}(\boldsymbol{x}_0)$. This obviously yields a lower bound once $f^\sharp \in \mathcal{F}$. If $\mathcal{F}$ includes only $f^\sharp$, then the bound is exact, but is computationally prohibitive due to the difficulty of optimizing $\boldsymbol{\delta}$. The idea is then to choose $\mathcal{F}$ properly to incorporate rich information of $f^\sharp$, while allowing us to calculate the lower bound in Eq.3 computationally tractably. In this paper, we consider the set of all functions bounded in $[0, 1]$, namely

$$\mathcal{F}_{[0,1]} = \left\{ f : f(\boldsymbol{z}) \in [0, 1], \forall \boldsymbol{z} \in \mathbb{R}^d \right\}, \tag{4}$$

which guarantees to include all $f^\sharp$ by definition.

Denote by $\mathcal{L}_{\pi_0}(\mathcal{F}, \mathcal{B})$ the lower bound in Eq.3. We can rewrite it into the following minimax form using the Lagrangian function,

$$\mathcal{L}_{\pi_0}(\mathcal{F}, \mathcal{B}) = \min_{f \in \mathcal{F}} \min_{\boldsymbol{\delta} \in \mathcal{B}} \max_{\lambda \in \mathbb{R}} L(f, \boldsymbol{\delta}, \lambda) \triangleq \min_{f \in \mathcal{F}} \min_{\boldsymbol{\delta} \in \mathcal{B}} \max_{\lambda \in \mathbb{R}} \left\{ f_{\pi_0}(\boldsymbol{x}_0 + \boldsymbol{\delta}) - \lambda(f_{\pi_0}(\boldsymbol{x}_0) - f^\sharp_{\pi_0}(\boldsymbol{x}_0)) \right\}, \tag{5}$$

where $\lambda$ is the Lagrangian multiplier. Exchanging the min and max yields the following dual form.

**Theorem 1.** *1)* *(Dual Form)* *Denote by $\pi_\delta$ the distribution of $z + \delta$ when $z \sim \pi_0$. Assume $\mathcal{F}$ and $\mathcal{B}$ are compact set. We have the following lower bound of $\mathcal{L}_{\pi_0}(\mathcal{F}, \mathcal{B})$:*

$$\mathcal{L}_{\pi_0}(\mathcal{F}, \mathcal{B}) \geq \max_{\lambda \geq 0} \min_{f \in \mathcal{F}} \min_{\delta \in \mathcal{B}} L(f, \delta, \lambda) = \max_{\lambda \geq 0} \left\{ \lambda f_{\pi_0}^\sharp(x_0) - \max_{\delta \in \mathcal{B}} \mathbb{D}_\mathcal{F}(\lambda \pi_0 \parallel \pi_\delta) \right\}, \quad (6)$$

*where we define the discrepancy term $\mathbb{D}_\mathcal{F}(\lambda \pi_0 \parallel \pi_\delta)$ as*

$$\max_{f \in \mathcal{F}} \left\{ \lambda \mathbb{E}_{z \sim \pi_0}[f(x_0 + z)] - \mathbb{E}_{z \sim \pi_\delta}[f(x_0 + z)] \right\},$$

*which measures the difference of $\lambda \pi_0$ and $\pi_\delta$ by seeking the maximum discrepancy of the expectation for $f \in \mathcal{F}$. As we will show later, the bound in (6) is computationally tractable with proper $(\mathcal{F}, \mathcal{B}, \pi_0)$.*

*II) When $\mathcal{F} = \mathcal{F}_{[0,1]} := \{f \colon f(x) \in [0,1], \ x \in \mathbb{R}^d\}$, we have in particular*

$$\mathbb{D}_{\mathcal{F}_{[0,1]}}(\lambda \pi_0 \parallel \pi_\delta) = \int (\lambda \pi_0(z) - \pi_\delta(z))_+ \, dz,$$

*where $(t)_+ = \max(0, t)$. Furthermore, we have $0 \leq \mathbb{D}_{\mathcal{F}_{[0,1]}}(\lambda \pi_0 \parallel \pi_\delta) \leq \lambda$ for any $\pi_0$, $\pi_\delta$ and $\lambda > 0$. Note that $\mathbb{D}_{\mathcal{F}_{[0,1]}}(\lambda \pi_0 \parallel \pi_\delta)$ coincides with the total variation distance between $\pi_0$ and $\pi_\delta$ when $\lambda = 1$.*

*III) (Strong duality) Suppose $\mathcal{F} = \mathcal{F}_{[0,1]}$ and suppose that for any $\lambda \geq 0$, $\min_{\delta \in \mathcal{B}} \min_{f \in \mathcal{F}_{[0,1]}} L(f, \delta, \lambda) = \min_{f \in \mathcal{F}_{[0,1]}} L(f, \delta^*, \lambda)$, for some $\delta^* \in \mathcal{B}$, we have*

$$\mathcal{L}_{\pi_0}(\mathcal{F}, \mathcal{B}) = \max_{\lambda \geq 0} \min_{\delta \in \mathcal{B}} \min_{f \in \mathcal{F}} L(f, \delta, \lambda).$$

**Remark** We will show later that the proposed methods and the cases we study satisfy the condition in part III of the theorem and thus all the lower bounds of the proposed method are tight.

Proof is deferred to Appendix A.1. Although the lower bound in Eq.6 still involves an optimization on $\delta$ and $\lambda$, both of them are much easier than the original adversarial optimization in Eq.1. With proper choices of $\mathcal{F}$, $\mathcal{B}$ and $\pi_0$, the optimization of $\delta$ can be shown to provide simple closed-form solutions by exploiting the symmetry of $\mathcal{B}$, and the optimization of $\lambda$ is a very simple one-dimensional searching problem.

As corollaries of Theorem 1, we can exactly recover the bound derived by [14] and [9] under our functional optimization framework, different from their original Neyman-Pearson lemma approaches.

**Corollary 1.** *With Laplacian noise $\pi_0(\cdot) = \mathrm{Laplace}(\cdot; b)$, where $\mathrm{Laplace}(x; b) = \frac{1}{(2b)^d} \exp(-\frac{\|x\|_1}{b})$, $\ell_1$ adversarial setting $\mathcal{B} = \{\delta \colon \|\delta\|_1 \leq r\}$ and $\mathcal{F} = \mathcal{F}_{[0,1]}$, the lower bound in Eq.6 becomes*

$$\max_{\lambda \geq 0} \left\{ \lambda f_{\pi_0}^\sharp(x_0) - \max_{\|\delta\|_1 \leq r} \mathbb{D}_{\mathcal{F}_{[0,1]}}(\lambda \pi_0 \| \pi_\delta) \right\} = \begin{cases} 1 - e^{r/b}(1 - f_{\pi_0}^\sharp(x_0)), & \text{when} f_{\pi_0}^\sharp(x_0) \geq 1 - \frac{1}{2}e^{-r/b}, \\[2mm] \frac{1}{2} e^{-\frac{r}{b} - \log[2(1 - f_{\pi_0}^\sharp(x_0)]}, & \text{when} f_{\pi_0}^\sharp(x_0) < 1 - \frac{1}{2}e^{-r/b}. \end{cases} \quad (7)$$

Thus, with our previous explanation, we obtain $\mathcal{L}_{\pi_0}(\mathcal{F}, \mathcal{B}) \geq \frac{1}{2} \iff r \leq -b \log\left[2(1 - f_{\pi_0}^\sharp(x_0))\right]$, which is exactly the $\ell_1$ certification radius derived by [14]. See Appendix A.2 for proof details. For Gaussian noise setting which has been frequently adopted, we have

**Corollary 2.** *With isotropic Gaussian noise $\pi_0 = \mathcal{N}(0, \sigma^2 I_{d \times d})$, $\ell_2$ attack $\mathcal{B} = \{\delta \colon \|\delta\|_2 \leq r\}$ and $\mathcal{F} = \mathcal{F}_{[0,1]}$, the lower bound in Eq.6 becomes*

$$\max_{\lambda \geq 0} \left\{ \lambda f_{\pi_0}^\sharp(x_0) - \max_{\|\delta\|_2 \leq r} \mathbb{D}_{\mathcal{F}_{[0,1]}}(\lambda \pi_0 \| \pi_\delta) \right\} = \Phi\left( \Phi^{-1}(f_{\pi_0}^\sharp(x_0)) - \frac{r}{\sigma} \right). \quad (8)$$

Analogously, we can retrieve the main theoretical result of [9] : $\mathcal{L}_{\pi_0}(\mathcal{F}, \mathcal{B}) \geq \frac{1}{2} \iff r \leq \sigma \Phi^{-1}(f_{\pi_0}^\sharp(x_0))$. See Appendix A.3 for proof details.

## 3.3 Trade-off Between Accuracy and Robustness

The lower bound in Eq.6 reflects an intuitive trade-off between the robustness and accuracy on the certification problem:

$$\max_{\lambda \geq 0} \left[ \lambda \underbrace{f_{\pi_0}^{\sharp}(\boldsymbol{x}_0)}_{\text{Accuracy}} + \underbrace{\left( -\max_{\boldsymbol{\delta} \in \mathcal{B}} \mathbb{D}_{\mathcal{F}} \left( \lambda \pi_0 \parallel \pi_{\boldsymbol{\delta}} \right) \right)}_{\text{Robustness}} \right], \tag{9}$$

where the first term reflects the accuracy of the smoothed classifier (assuming the true label is $y = 1$), while the second term $-\max_{\boldsymbol{\delta} \in \mathcal{B}} \mathbb{D}_{\mathcal{F}} \left( \lambda \pi_0 \parallel \pi_{\boldsymbol{\delta}} \right)$ measures the robustness of the smoothing method, via the negative maximum discrepancy between the original smoothing distribution $\pi_0$ and perturbed distribution $\pi_{\boldsymbol{\delta}}$ for $\boldsymbol{\delta} \in \mathcal{B}$. The maximization of dual coefficient $\lambda$ can be viewed as searching for a best balance between these two terms to achieve the largest lower bound.

More critically, different choices of smoothing distributions yields a trade-off between accuracy and robustness in Eq.9. A good choice of the smoothing distribution should ① be centripetal enough to obtain a large $f_{\pi_0}^{\sharp}(\boldsymbol{x}_0)$ and ② have large kurtosis or long tail to yield a small $\max_{\boldsymbol{\delta} \in \mathcal{B}} \mathbb{D}_{\mathcal{F}}(\lambda \pi_0 \parallel \pi_{\boldsymbol{\delta}})$ discrepancy term. In the next section, we'll show how to design a distribution that could improve both points.

# 4 Improving Certification Bounds with a New Distribution Family

## 4.1 *"Thin Shell"* Phenomenon and New Distribution Family

We first identify a key problem of the usage of Laplacian and Gaussian noise in high dimensional space, due to the "*thin shell*" phenomenon that the probability mass of them concentrates on a sphere far away from the center points [20].

**Proposition 1** ([20], Section 3.1). *Let $\boldsymbol{z} \sim \mathcal{N}(\boldsymbol{0}, I_{d \times d})$ be a $d$-dimensional standard Gaussian random variable. Then there exists a constant $c$, such that for any $\delta \in (0, 1)$, Prob $\left( \sqrt{d} - \sqrt{c \log(2/\delta)} \leq \|\boldsymbol{z}\|_2 \leq \sqrt{d} + \sqrt{c \log(2/\delta)} \right) \geq 1 - \delta$. See [20] for more discussion.*

This suggests that with high probability, $\boldsymbol{z}$ takes values very close to the sphere of radius $\sqrt{d}$, within a constant distance from that sphere. There exists similar phenomenon for Laplacian distribution:

**Proposition 2** (Chebyshev bound). *Let $\boldsymbol{z}$ be a $d$-dimensional Laplacian random variable, $\boldsymbol{z} = (z_1, \cdots, z_d)$, where $z_i \sim \text{Laplace}(1), i = 1, \cdots, d$. Then for any $\delta \in (0, 1)$, we have Prob $\left( 1 - 1/\sqrt{d\delta} \leq \|\boldsymbol{z}\|_1 / d \leq 1 + 1/\sqrt{d\delta} \right) \geq 1 - \delta$.*

Although choosing isotropic Laplacian and Gaussian distribution appears to be natural, this "*thin shell*" phenomenon makes it sub-optimal to use them for adversarial certification, because one would expect that the smoothing distribution should concentrate around the center (the original image) in order to make the smoothed classifier accurate enough in trade-off of Eq.9.

Thus it's desirable to design a distribution more *concentrated* to center. To motivate our new distribution family, it's useful to examine the density function of the distributions of the radius of spherical distributions in general.

**Proposition 3.** *Assume $\boldsymbol{z}$ is a symmetric random variable on $\mathbb{R}^d$ with a probability density function (PDF) of form $\pi_0(\boldsymbol{z}) \propto \phi(\|\boldsymbol{z}\|)$, where $\phi \colon [0, \infty) \to [0, \infty)$ is a univariate function, then the PDF of the norm of $\boldsymbol{z}$ is $p_{\|\boldsymbol{z}\|}(r) \propto r^{d-1}\phi(r)$. The term $r^{d-1}$ arises due to the integration on the surface of radius $r$ norm ball in $\mathbb{R}^d$. Here $\|\cdot\|$ can be any $L_p$ norm.*

In particular, for $\boldsymbol{z} \sim \mathcal{N}(0, \sigma^2 I_{d \times d})$, we have $\phi(r) \propto \exp(-r^2/(2\sigma^2))$ and hence $p_{\|\boldsymbol{z}\|_2}(r) \propto r^{d-1} \exp(-r^2/(2\sigma^2))$. We can see that the "*thin shell*" phenomenon is caused by the $r^{d-1}$ term, which makes the density to be highly peaked when $d$ is large. To alleviate the concentration phenomenon, we need to cancel out the effect of $r^{d-1}$, which motivates the following family of smoothing distributions:

$$\pi_0(\boldsymbol{z}) \propto \|\boldsymbol{z}\|_{n_1}^{-k} \exp\left( -\frac{\|\boldsymbol{z}\|_{n_2}^p}{b} \right),$$

where parameters $k, n_1, n_2, p \in \mathbb{N}$. Next we discuss how to choose suitable parameters depending on specific perturbation region.

## 4.2 $\ell_1$ and $\ell_2$ Region Certification

Based on original Laplacian and Gaussian distributions and above intuition, we propose:

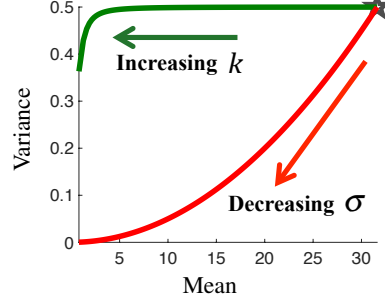

$$\ell_1 : \pi_{\mathbf{0}}(\mathbf{z}) \propto \|\mathbf{z}\|_1^{-k} \exp\left(-\frac{\|\mathbf{z}\|_1}{b}\right) \qquad (10)$$

$$\ell_2 : \pi_{\mathbf{0}}(\mathbf{z}) \propto \|\mathbf{z}\|_2^{-k} \exp\left(-\frac{\|\mathbf{z}\|_2^2}{2\sigma^2}\right) \qquad (11)$$

where we introduce the $\|\mathbf{z}\|^{-k}$ term in $\pi_{\mathbf{0}}$, with $k$ a positive parameter, to make the radius distribution more concentrated when $k$ is large.

The radius distribution in Eq.10 and Eq.11 is controlled by two parameters: $\sigma$ (or $b$) and $k$, who control the scale and shape of the distribution, respectively. The key idea is that adjusting extra parameter $k$ allows us to control the trade-off the accuracy and robustness more precisely. As shown in Fig.1, adjusting $\sigma$ moves the mean close to zero

Figure 1: Starting from radius distribution in Eq.11 with $d = 100$ $\sigma = 1$ and $k = 0$ (black start), increasing $k$ (green curve) moves the mean towards zero *without significantly reducing the variance*. Decreasing $\sigma$ (red curve) can also decrease the mean, but with a cost of decreasing the variance quadratically.

(hence ① yielding higher accuracy), but at cost of decreasing the variance quadratically (hence ② less robust). In contrast, adjusting $k$ decreases the mean without significantly impacting the variance, thus yield a much better trade-off on accuracy and robustness.

**Computational Method** Now we no longer have the closed-form solution of the bound like Eq.7 and Eq.8. However, efficient computational methods can still be developed for calculating the bound in Eq.6 with $\pi_{\mathbf{0}}$ in Eq.11 or Eq.11. The key is that the maximum of the distance term $\mathbb{D}_{\mathcal{F}_{[0,1]}}(\lambda\pi_{\mathbf{0}} \| \pi_{\boldsymbol{\delta}})$ over $\boldsymbol{\delta} \in \mathcal{B}$ is always achieved on the boundary of $\mathcal{B}$:

**Theorem 2.** *Consider the $\ell_1$ attack with $\mathcal{B} = \{\boldsymbol{\delta} : \|\boldsymbol{\delta}\|_1 \le r\}$ and smoothing distribution $\pi_{\mathbf{0}}(\mathbf{z}) \propto \|\mathbf{z}\|_1^{-k} \exp\left(-\frac{\|\mathbf{z}\|_1}{b}\right)$ with $k \ge 0$ and $b > 0$, or the $\ell_2$ attack with $\mathcal{B} = \{\boldsymbol{\delta} : \|\boldsymbol{\delta}\|_2 \le r\}$ and smoothing distribution $\pi_{\mathbf{0}}(\mathbf{z}) \propto \|\mathbf{z}\|_2^{-k} \exp\left(-\frac{\|\mathbf{z}\|_2^2}{2\sigma^2}\right)$ with $k \ge 0$ and $\sigma > 0$. Define $\boldsymbol{\delta}^* = [r, 0, ..., 0]^\top$, we have*

$$\mathbb{D}_{\mathcal{F}_{[0,1]}}(\lambda\pi_{\mathbf{0}} \| \pi_{\boldsymbol{\delta}^*}) = \max_{\boldsymbol{\delta} \in \mathcal{B}} \mathbb{D}_{\mathcal{F}_{[0,1]}}(\lambda\pi_{\mathbf{0}} \| \pi_{\boldsymbol{\delta}})$$

*for any positive $\lambda$.*

With Theorem 2, we can compute Eq.6 with $\boldsymbol{\delta} = \boldsymbol{\delta}^*$. We then calculate $\mathbb{D}_{\mathcal{F}_{[0,1]}}(\lambda\pi_{\mathbf{0}} \| \pi_{\boldsymbol{\delta}^*}) = \mathbb{E}_{\mathbf{z} \sim \pi_{\mathbf{0}}}\left[\left(\lambda - \frac{\pi_{\boldsymbol{\delta}^*}(\mathbf{z})}{\pi_{\mathbf{0}}(\mathbf{z})}\right)_+\right]$ using Monte Carlo approximation with i.i.d. samples $\{\mathbf{z}_i\}_{i=1}^n$ be i.i.d. samples from $\pi_{\mathbf{0}}$: $\hat{D} := \frac{1}{n}\sum_{i=1}^n (\lambda - \pi_{\boldsymbol{\delta}^*}(\mathbf{z}_i)/\pi_{\mathbf{0}}(\mathbf{z}_i))_+$, which is bounded in the following confidence interval $[\hat{D} - \lambda\sqrt{\log(2/\delta)/(2n)}, \hat{D} + \lambda\sqrt{\log(2/\delta)/(2n)}]$ with confidence level $1 - \delta$ for $\delta \in (0, 1)$. What's more, the optimization on $\lambda \ge 0$ is one-dimensional and can be solved numerically efficiently (see Appendix for details).

## 4.3 $\ell_\infty$ Region Certification

Going further, we consider the more difficult $\ell_\infty$ attack whose attacking region is $\mathcal{B}_{\ell_\infty, r} = \{\boldsymbol{\delta} : \|\boldsymbol{\delta}\|_\infty \le r\}$. The commonly used Gaussian smoothing distribution, as well as our $\ell_2$-based smoothing distribution in Eq.11, is unsuitable for this region:

**Proposition 4.** *With the smoothing distribution $\pi_{\mathbf{0}}$ in Eq.11 for $k \ge 0, \sigma > 0$, and $\mathcal{F} = \mathcal{F}_{[0,1]}$ shown in Eq.4, the bound we get for certifying the $\ell_\infty$ attack on $\mathcal{B}_{\ell_\infty, r} = \{\boldsymbol{\delta} : \|\boldsymbol{\delta}\|_\infty \le r\}$ is equivalent to that for certifying the $\ell_2$ attack on $\mathcal{B}_{\ell_2, \sqrt{d}r} = \{\boldsymbol{\delta} : \|\boldsymbol{\delta}\|_2 \le \sqrt{d}r\}$, that is,*

$$\mathcal{L}_{\pi_{\mathbf{0}}}(\mathcal{F}_{[0,1]}, \mathcal{B}_{\ell_\infty, r}) = \mathcal{L}_{\pi_{\mathbf{0}}}(\mathcal{F}_{[0,1]}, \mathcal{B}_{\ell_2, \sqrt{d}r}).$$

As shown in this proposition, if we use $\ell_2$ distribution in Eq.11 for certification, the bound we obtain is effectively the bound we get for verifying a $\ell_2$ ball with radius $\sqrt{d}r$, which is too large to give meaningful results due to high dimension.

In order to address this problem, we extend our proposed distribution family with new distributions which are more suitable for $\ell_\infty$ certification setting:

$$\pi_{\mathbf{0}}(\mathbf{z}) \propto \|\mathbf{z}\|_\infty^{-k} \exp\left(-\frac{\|\mathbf{z}\|_\infty^2}{2\sigma^2}\right), \qquad (12)$$

$$\pi_{\mathbf{0}}(\mathbf{z}) \propto \|\mathbf{z}\|_\infty^{-k} \exp\left(-\frac{\|\mathbf{z}\|_2^2}{2\sigma^2}\right). \qquad (13)$$

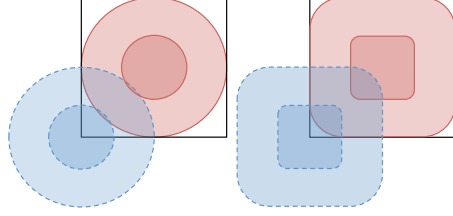

Figure 2: For $\ell_\infty$ attacking, compared with the distribution in Eq.11, the mixed norm distribution in Eq.13 (right) yields smaller discrepancy term (because of larger overlap areas), and hence higher robustness and better confidence bound. The distribution described in Eq.12 has the same impact.

The motivation is to allocate more probability mass along the "pointy" directions with larger $\ell_\infty$ norm, and hence decrease the maximum discrepancy term $\max_{\boldsymbol{\delta} \in \mathcal{B}_{\ell_\infty,r}} \mathbb{D}_{\mathcal{F}}(\lambda\pi_{\mathbf{0}} \parallel \pi_{\boldsymbol{\delta}})$, see Fig.2.

**Computational Method**  In order to compute the lower bound with proposed distribution, we need to establish similar theoretical results as Theorem 2, showing the optimal $\boldsymbol{\delta}$ is achieved at one vertex (the pointy points) of $\ell_\infty$ ball.

**Theorem 3.** *Consider the $\ell_\infty$ attack with $\mathcal{B}_{\ell_\infty,r} = \{\boldsymbol{\delta} : \|\boldsymbol{\delta}\|_\infty \leq r\}$ and the mixed norm smoothing distribution in Eq.13 with $k \geq 0$ and $\sigma > 0$. Define $\boldsymbol{\delta}^* = [r, r, ..., r]^\top$. We have for any $\lambda > 0$,*

$$\mathbb{D}_{\mathcal{F}_{[0,1]}}\left(\lambda\pi_{\mathbf{0}} \parallel \pi_{\boldsymbol{\delta}^*}\right) = \max_{\delta \in \mathcal{B}} \mathbb{D}_{\mathcal{F}_{[0,1]}}\left(\lambda\pi_{\mathbf{0}} \parallel \pi_{\boldsymbol{\delta}}\right).$$

The proofs of Theorem 2 and 3 are non-trivial and deferred to Appendix. With the optimal $\boldsymbol{\delta}^*$ found above, we can calculate the bound with similar Monte Carlo approximation outlined in Section 4.2.

## 5  Experiments

We evaluate proposed certification bound and smoothing distributions for $\ell_1$, $\ell_2$ and $\ell_\infty$ attacks. We compare with the randomized smoothing method of [14] with Laplacian smoothing for $\ell_1$ region cerification. For $\ell_2$ and $\ell_\infty$ cases, we regard the method derived by [9] with Gaussian smoothing distribution as the baseline. For fair comparisons, we use the same model architecture and pretrained models provided by [14], [9] and [10], which are ResNet-110 for CIFAR-10 and ResNet-50 for ImageNet. We use the official code[2] provided by [9] for all the following experiments. For all other details and parameter settings, we refer the readers to Appendix B.2.

| $\ell_1$ RADIUS (CIFAR-10) | 0.25 | 0.5 | 0.75 | 1.0 | 1.25 | 1.5 | 1.75 | 2.0 | 2.25 |
|---|---|---|---|---|---|---|---|---|---|
| BASELINE (%) | 62 | 49 | 38 | 30 | 23 | 19 | 17 | 14 | 12 |
| OURS (%) | **64** | **51** | **41** | **34** | **27** | **22** | **18** | **17** | **14** |

| $\ell_1$ RADIUS (IMAGENET) | 0.5 | 1.0 | 1.5 | 2.0 | 2.5 | 3.0 | 3.5 |
|---|---|---|---|---|---|---|---|
| BASELINE (%) | 50 | 41 | 33 | 29 | 25 | 18 | 15 |
| OURS (%) | **51** | **42** | **36** | **30** | **26** | **22** | **16** |

Table 1: Certified top-1 accuracy of the best classifiers with various $\ell_1$ radius.

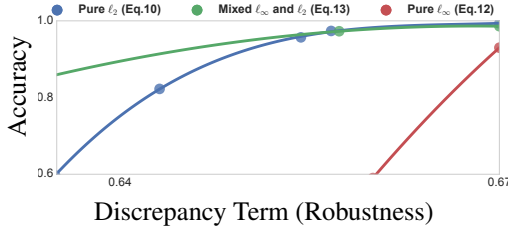
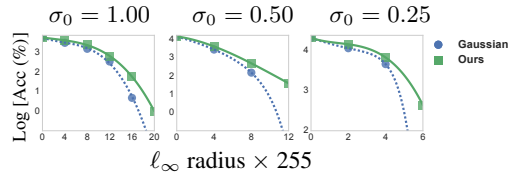

Figure 3: The Pareto frontier of accuracy and robustness (in the sense of Eq.9) of the three smoothing families in Eq.11, Eq.13, and Eq.12 for $\ell_\infty$ attacking, when we search for the best parameters $(k, \sigma)$ for each of them. The mixed norm family Eq.13 yields the best trade-off than the other two. We assume $f^\sharp(\boldsymbol{x}) = \mathbb{I}(\|\boldsymbol{x}\|_2 \leq r)$ and dimension $d = 5$. The case when $f^\sharp(\boldsymbol{x}) = \mathbb{I}(\|\boldsymbol{x}\|_\infty \leq r)$ has similar result (not shown).

Figure 4: Results of $\ell_\infty$ verification on CIFAR-10, on models trained with Gaussian noise data augmentation with different variances $\sigma_0$. Our method obtains consistently better results.

**Evaluation Metrics** Methods are evaluated with the certified accuracy defined in [9]. Given an input image $\boldsymbol{x}$ and a perturbation region $\mathcal{B}$, the smoothed classifier certifies image $\boldsymbol{x}$ correctly if the prediction is correct and has a guaranteed confidence lower bound larger than $1/2$ for any $\boldsymbol{\delta} \in \mathcal{B}$. The certified accuracy is the percentage of images that are certified correctly. Following [10], we calculate the certified accuracy of all the classifiers in [9] or [10] for various radius, and report the best results over all of classifiers.

## 5.1 $\ell_1$ & $\ell_2$ Certification

For $\ell_1$ certification, we compare our method with [14] on CIFAR-10 and ImageNet with the type 1 trained model in [14]. As shown in Table 1, our non-Laplacian centripetal distribution consistently outperforms the result of baseline for any $\ell_1$ radius.

| $\ell_2$ RADIUS (CIFAR-10) | 0.25 | 0.5 | 0.75 | 1.0 | 1.25 | 1.5 | 1.75 | 2.0 | 2.25 |
|---|---|---|---|---|---|---|---|---|---|
| BASELINE (%) | 60 | 43 | 34 | 23 | 17 | 14 | 12 | 10 | 8 |
| OURS (%) | **61** | **46** | **37** | **25** | **19** | **16** | **14** | **11** | **9** |

| $\ell_2$ RADIUS (IMAGENET) | 0.5 | 1.0 | 1.5 | 2.0 | 2.5 | 3.0 | 3.5 |
|---|---|---|---|---|---|---|---|
| BASELINE (%) | 49 | 37 | 29 | 19 | 15 | 12 | 9 |
| OURS (%) | **50** | **39** | **31** | **21** | **17** | **13** | **10** |

Table 2: Certified top-1 accuracy of the best classifiers with various $\ell_2$ radius.

For $\ell_2$ certification, we compare our method with [9] on CIFAR-10 and ImageNet. For a fair comparison, we use the same pre-trained models as [9], which is trained with Gaussian noise on both CIFAR-10 and ImageNet dataset. Table 2 reports the certified accuracy of our method and the baseline on CIFAR-10 and ImageNet . We find that our method consistently outperforms the baseline. The readers are referred to the Appendix B.3 for detailed ablation studies.

## 5.2 $\ell_\infty$ Certification

**Toy Example** We first construct a simple toy example to verify the advantages of the distribution Eq.13 and Eq.12 over the $\ell_2$ family in Eq.11. We set the true classifier to be $\tilde{f}^\sharp(\boldsymbol{x}) = \mathbb{I}(\|x\|_2 \leq r)$ in $r = 0.65$, $d = 5$ case and plot in Fig.3 the Pareto frontier of the accuracy and robustness terms in Eq.9 for the three families of smoothing distributions, as we search for the best combinations of parameters $(k, \sigma)$. The mixed norm smoothing distribution clearly obtain the best trade-off on accuracy and robustness, and hence guarantees a tighter lower bound for certification. Fig.3 also shows that Eq.12 even performs worse than Eq.11. We further theoretically show that Eq.12 is provably not suitable for $\ell_\infty$ region certification in Appendix A.5.

**CIFAR-10**  Based on above results, we only compared the method defined by Eq.13 with [10] on CIFAR-10. The certified accuracy of our method and the baseline using Gaussian smoothing distribution and Proposition 4 are shown in Table 3. We can see that our method consistently outperforms the Gaussian baseline by a large margin. More clarification about $\ell_\infty$ experiments is in Appendix **??**.

| $l_\infty$ RADIUS | 2/255 | 4/255 | 6/255 | 8/255 | 10/255 | 12/255 |
|---|---|---|---|---|---|---|
| BASELINE (%) | 58 | 42 | 31 | 25 | 18 | 13 |
| OURS (%) | **60** | **47** | **38** | **32** | **23** | **17** |

Table 3: Certified top-1 accuracy of the best classifiers with various $l_\infty$ radius on CIFAR-10.

To further confirm the advantage of our method, we plot in Fig.4 the certified accuracy of our method and Gaussian baseline using models trained with Gaussian perturbation of different variances $\sigma_0$ under different $\ell_\infty$ radius. Our approach outperforms baseline consistently, especially when the $\ell_\infty$ radius is large. We also experimented our method and baseline on ImageNet but did not obtain non-trivial results. This is because $\ell_\infty$ verification is extremely hard with very large dimensions [32, 31]. Future work will investigate how to obtain non-trivial bounds for $\ell_\infty$ attacking at ImageNet scales with smoothing classifiers.

# 6  Conclusion

We propose a general functional optimization based framework of adversarial certification with non-Gaussian smoothing distributions. Based on the insights from our new framework and high dimensional geometry, we propose a new family of non-Gaussian smoothing distributions, which outperform the Gaussian and Laplace smoothing for certifying $\ell_1$, $\ell_2$ and $\ell_\infty$ attacking. Our work provides a basis for a variety of future directions, including improved methods for $\ell_p$ attacks, and tighter bounds based on adding additional constraints to our optimization framework.

## Broader Impact

Adversarial certification via randomized smoothing could achieve *guaranteed* robust machine learning models, thus has wide application on AI security. a & b) With our empirical results, security engineers could get better performance on defending against vicious attacks; With our theoretical results, it will be easier for following researchers to derive new bounds for different kinds of smoothing methods. We don't foresee the possibility that it could bring negative social impacts. c) Our framework is mathematically rigorous thus would never fail. d) Our method doesn't have bias in data as we provide a general certification method for all tasks and data, and our distribution is not adaptive towards data.

## Acknowledgement

Gong, Ye, Liu are supported in part by NSF CAREER 1846421. Zhu is supported in part by Beijing Nova Program (No. 202072) from Beijing Municipal Science & Technology Commission. We would like to thank Tongzheng Ren, Jiaye Teng, Yang Yuan and the reviewers for helpful suggestions that improved the paper.

## Footnotes

[2]https://github.com/locuslab/smoothing. Our results are slightly different with those in original paper due to the randomness of sampling.

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
