[Supplementary Material]

# A Proofs

## A.1 Proof for Theorem 1

### A.1.1 Proof for (I) and (II)

First, observe that the constraint in Equation (3) can be equivalently replaced by an inequality constraint $f_{\pi_0}(\boldsymbol{x}_0) \geq f^\sharp_{\pi_0}(\boldsymbol{x}_0)$. Therefore, the Lagrangian multiplier can be restricted to be $\lambda \geq 0$. We have

$$
\begin{aligned}
\mathcal{L}_{\pi_0}(\mathcal{F}, \mathcal{B}) &= \min_{\boldsymbol{\delta} \in \mathcal{B}} \min_{f \in \mathcal{F}} \max_{\lambda \geq 0} \mathbb{E}_{\pi_\delta}[f(\boldsymbol{x}_0 + \boldsymbol{z})] + \lambda \left( f^\sharp_{\pi_0}(\boldsymbol{x}_0) - \mathbb{E}_{\pi_0}[f(\boldsymbol{x}_0 + \boldsymbol{z})] \right) \\
&\geq \max_{\lambda \geq 0} \min_{\boldsymbol{\delta} \in \mathcal{B}} \min_{f \in \mathcal{F}} \mathbb{E}_{\pi_\delta}[f(\boldsymbol{x}_0 + \boldsymbol{z})] + \lambda \left( f^\sharp_{\pi_0}(\boldsymbol{x}_0) - \mathbb{E}_{\pi_0}[f(\boldsymbol{x}_0 + \boldsymbol{z})] \right) \\
&= \max_{\lambda \geq 0} \min_{\boldsymbol{\delta} \in \mathcal{B}} \left\{ \lambda f^\sharp_{\pi_0}(\boldsymbol{x}_0) + \min_{f \in \mathcal{F}} \mathbb{E}_{\pi_\delta}[f(\boldsymbol{x}_0 + \boldsymbol{z})] - \lambda \mathbb{E}_{\pi_0}[f(\boldsymbol{x}_0 + \boldsymbol{z})]) \right\} \\
&= \max_{\lambda \geq 0} \min_{\boldsymbol{\delta} \in \mathcal{B}} \left\{ \lambda f^\sharp_{\pi_0}(\boldsymbol{x}_0) - \mathbb{D}_{\mathcal{F}}(\lambda \pi_0 \| \pi_\delta) \right\}.
\end{aligned}
$$

II) follows a straightforward calculation.

### A.1.2 Proof for (III), the strong duality

We first introduce the following lemma, which is a straight forward generalization of the strong Lagrange duality to functional optimization case.

**Lemma 1.** *Given some $\boldsymbol{\delta}^*$, we have*

$$
\begin{aligned}
\max_{\lambda \in \mathbb{R}} \min_{f \in \mathcal{F}_{[0,1]}} & \mathbb{E}_{\pi_{\delta^*}} \left[ f(\boldsymbol{x}_0 + \boldsymbol{z}) \right] + \lambda \left( f^\sharp_{\pi_0}(\boldsymbol{x}_0) - \mathbb{E}_{\pi_0} \left[ f(\boldsymbol{x}_0 + \boldsymbol{z}) \right] \right) \\
= \min_{f \in \mathcal{F}_{[0,1]}} \max_{\lambda \in \mathbb{R}} & \mathbb{E}_{\pi_{\delta^*}} \left[ f(\boldsymbol{x}_0 + \boldsymbol{z}) \right] + \lambda \left( f^\sharp_{\pi_0}(\boldsymbol{x}_0) - \mathbb{E}_{\pi_0} \left[ f(\boldsymbol{x}_0 + \boldsymbol{z}) \right] \right).
\end{aligned}
$$

The proof of Lemma 1 is standard. However, for completeness, we include it here.

*Proof.* Without loss of generality, we assume $f^\sharp_{\pi_0}(\boldsymbol{x}_0) \in (0, 1)$, otherwise the feasible set is trivial.

Let $\alpha^*$ be the value of the optimal solution of the primal problem. We define $f^\sharp_{\pi_0}(\boldsymbol{x}_0) - \mathbb{E}_{\pi_0} \left[ f(\boldsymbol{x}_0 + \boldsymbol{z}) \right] = h[f]$ and $g[f] = \mathbb{E}_{\pi_{\delta^*}} \left[ f(\boldsymbol{x}_0 + \boldsymbol{z}) \right]$. We define the following two sets:

$$
\begin{aligned}
\mathcal{A} &= \left\{ (v, t) \in \mathbb{R} \times \mathbb{R} : \exists f \in \mathcal{F}_{[0,1]}, h[f] = v, g[f] \leq t \right\} \\
\mathcal{B} &= \left\{ (0, s) \in \mathbb{R} \times \mathbb{R} : s < \alpha^* \right\}.
\end{aligned}
$$

Notice that both sets $\mathcal{A}$ and $\mathcal{B}$ are convex. This is obvious for $\mathcal{B}$. For any $(v_1, t_1) \in \mathcal{A}$ and $(v_2, t_2) \in \mathcal{A}$, we define $f_1 \in \mathcal{F}_{[0,1]}$ such that $h[f_1] = v_1, g[f_1] \leq t_1$ (and similarly we define $f_2$). Notice that for any $\gamma \in [0, 1]$, we have

$$
\begin{aligned}
\gamma f_1 + (1 - \gamma) f_2 &\in \mathcal{F}_{[0,1]} \\
\gamma h[f_1] + (1 - \gamma) h[f_2] &= \gamma v_1 + (1 - \gamma) v_2 \\
\gamma g[f_1] + (1 - \gamma) g[f_2] &\leq \gamma t_1 + (1 - \gamma) t_2,
\end{aligned}
$$

which implies that $\gamma(v_1, t_1) + (1 - \gamma)(v_2, t_2) \in \mathcal{A}$ and thus $\mathcal{A}$ is convex. Also notice that by definition, $\mathcal{A} \cap \mathcal{B} = \emptyset$. Using separating hyperplane theorem, there exists a point $(q_1, q_2) \neq (0, 0)$ and a value $\alpha$ such that for any $(v, t) \in \mathcal{A}$, $q_1 v + q_2 t \geq \alpha$ and for any $(0, s) \in \mathcal{B}$, $q_2 s \leq \alpha$. Notice that we must have $q_2 \geq 0$, otherwise, for sufficient $s$, we will have $q_2 s > \alpha$. We thus have, for any $f \in \mathcal{F}_{[0,1]}$, we have

$$
q_1 h[f] + q_2 g[f] \geq \alpha^* \geq q_2 \alpha^*.
$$

If $q_2 > 0$, we have

$$
\max_{\lambda \in \mathbb{R}} \min_{f \in \mathcal{F}_{[0,1]}} g[f] + \lambda h[f] \geq \min_{f \in \mathcal{F}_{[0,1]}} g[f] + \frac{q_1}{q_2} h[f] \geq \alpha^*,
$$

which gives the strong duality. If $q_2 = 0$, we have for any $f \in \mathcal{F}_{[0,1]}$, $q_1 h[f] \geq 0$ and by the separating hyperplane theorem, $q_1 \neq 0$. However, this case is impossible: If $q_1 > 0$, choosing

$f \equiv 1$ gives $q_1 h[f] = q_1 \left(f_{\pi_0}^\sharp(x_0) - 1\right) < 0$; If $q_1 < 0$, by choosing $f \equiv 0$, we have $q_1 h[f] = q_1 \left(f_{\pi_0}^\sharp(x_0) - 0\right) < 0$. Both cases give contradiction.

$\square$

Based on Lemma 1, we have the proof of the strong duality as follows.

Notice that by Lagrange multiplier method, our primal problem can be rewritten as follows:

$$\min_{\delta \in \mathcal{B}} \min_{f \in \mathcal{F}_{[0,1]}} \max_{\lambda \in \mathbb{R}} \mathbb{E}_{\pi_\delta}\left[f(x_0 + z)\right] + \lambda \left(f_{\pi_0}^\sharp(x_0) - \mathbb{E}_{\pi_0}\left[f(x_0 + z)\right]\right),$$

and the dual problem is

$$\max_{\lambda \in \mathbb{R}} \min_{\delta \in \mathcal{B}} \min_{f \in \mathcal{F}_{[0,1]}} \mathbb{E}_{\pi_\delta}\left[f(x_0 + z)\right] + \lambda \left(f_{\pi_0}^\sharp(x_0) - \mathbb{E}_{\pi_0}\left[f(x_0 + z)\right]\right)$$

$$= \max_{\lambda \geq 0} \min_{\delta \in \mathcal{B}} \min_{f \in \mathcal{F}_{[0,1]}} \mathbb{E}_{\pi_\delta}\left[f(x_0 + z)\right] + \lambda \left(f_{\pi_0}^\sharp(x_0) - \mathbb{E}_{\pi_0}\left[f(x_0 + z)\right]\right).$$

By the assumption that for any $\lambda \geq 0$, we have

$$\max_{\lambda \geq 0} \min_{\delta \in \mathcal{B}} \min_{f \in \mathcal{F}_{[0,1]}} \mathbb{E}_{\pi_\delta}\left[f(x_0 + z)\right] + \lambda \left(f_{\pi_0}^\sharp(x_0) - \mathbb{E}_{\pi_0}\left[f(x_0 + z)\right]\right)$$

$$= \max_{\lambda \geq 0} \min_{f \in \mathcal{F}_{[0,1]}} \mathbb{E}_{\pi_{\delta^*}}\left[f(x_0 + z)\right] + \lambda \left(f_{\pi_0}^\sharp(x_0) - \mathbb{E}_{\pi_0}\left[f(x_0 + z)\right]\right),$$

for some $\delta^* \in \mathcal{B}$. We have

$$\max_{\lambda \in \mathbb{R}} \min_{\delta \in \mathcal{B}} \min_{f \in \mathcal{F}_{[0,1]}} \mathbb{E}_{\pi_\delta}\left[f(x_0 + z)\right] + \lambda \left(f_{\pi_0}^\sharp(x_0) - \mathbb{E}_{\pi_0}\left[f(x_0 + z)\right]\right)$$

$$= \max_{\lambda \geq 0} \min_{f \in \mathcal{F}_{[0,1]}} \mathbb{E}_{\pi_{\delta^*}}\left[f(x_0 + z)\right] + \lambda \left(f_{\pi_0}^\sharp(x_0) - \mathbb{E}_{\pi_0}\left[f(x_0 + z)\right]\right)$$

$$= \max_{\lambda \in \mathbb{R}} \min_{f \in \mathcal{F}_{[0,1]}} \mathbb{E}_{\pi_{\delta^*}}\left[f(x_0 + z)\right] + \lambda \left(f_{\pi_0}^\sharp(x_0) - \mathbb{E}_{\pi_0}\left[f(x_0 + z)\right]\right)$$

$$\overset{*}{=} \min_{f \in \mathcal{F}_{[0,1]}} \max_{\lambda \in \mathbb{R}} \mathbb{E}_{\pi_{\delta^*}}\left[f(x_0 + z)\right] + \lambda \left(f_{\pi_0}^\sharp(x_0) - \mathbb{E}_{\pi_0}\left[f(x_0 + z)\right]\right)$$

$$\geq \min_{\delta \in \mathcal{B}} \min_{f \in \mathcal{F}_{[0,1]}} \max_{\lambda \in \mathbb{R}} \mathbb{E}_{\pi_{\delta^*}}\left[f(x_0 + z)\right] + \lambda \left(f_{\pi_0}^\sharp(x_0) - \mathbb{E}_{\pi_0}\left[f(x_0 + z)\right]\right),$$

where the second equality (*) is by Lemma 1.

## A.2 Proof for Corollary 1

*Proof.* Given our confidence lower bound

$$\max_{\lambda \geq 0} \min_{\|\delta\|_1 \leq r} \left\{\lambda p_0 - \int \left(\lambda \pi_0(z) - \pi_\delta(z)\right)_+ dz\right\},$$

One can show that the worst case for $\delta$ is obtained when $\delta^* = (r, 0, \cdots, 0)$ (see following subsection), thus the bound is

$$\max_{\lambda \geq 0} \left\{\lambda p_0 - \int \frac{1}{2b} \exp\left(-\frac{|z_1|}{b}\right)\left[\lambda - \exp\left(\frac{|z_1| - |z_1 + r|}{b}\right)\right]_+ dz_1\right\}.$$

Denote $a$ to be the solution of $\lambda = \exp\left(\frac{|a| - |a + r|}{b}\right)$, then obviously we have

$$a = \begin{cases} -\infty, & b \log \lambda \geq r \\ -\frac{1}{2}\left(b \log \lambda + r\right), & -r < b \log \lambda < r \\ +\infty. & b \log \lambda \leq -r \end{cases}$$

So the bound above is

$$\lambda \int_{z_1 > a} \frac{1}{2b} \exp\left(-\frac{|z_1|}{b}\right) dz_1 - \int_{z_1 > a} \frac{1}{2b} \exp\left(-\frac{|z_1 + r|}{b}\right) dz_1.$$

i) $b \log \lambda \geq r \Leftrightarrow \lambda \geq \exp\left(\frac{r}{b}\right)$
the bound is
$$\max_{\lambda \geq e^{r/b}} \{\lambda p_0 - (\lambda - 1)\} = 1 - \exp\left(\frac{r}{b}\right)(1 - p_0).$$

ii) $-r < b \log \lambda < r \Leftrightarrow \exp\left(-\frac{r}{b}\right) < \lambda < \exp\left(\frac{r}{b}\right)$
the bound is
$$\max_{\lambda} \left\{\lambda p_0 - \lambda\left[1 - \frac{1}{2}\exp\left(-\frac{b \log \lambda + r}{2b}\right)\right] + \frac{1}{2}\exp\left(\frac{b \log \lambda - r}{2b}\right)\right\}$$
$$= \max_{\lambda} \left\{\lambda(p_0 - 1) + \frac{\lambda}{2}\exp\left(-\frac{b \log \lambda + r}{2b}\right) + \frac{1}{2}\exp\left(\frac{b \log \lambda - r}{2b}\right)\right\}$$
$$= \frac{1}{2}\exp\left(-\log\left[2(1 - p_0)\right] - \frac{r}{b}\right).$$

the extremum is achieved when $\hat{\lambda} = \exp\left(-2\log\left[2(1 - p_0)\right] - \frac{r}{b}\right)$. Notice that $\hat{\lambda}$ does not necessarily locate in $\left(e^{-r/b}, e^{r/b}\right)$, so the actual bound is always equal or less than $\frac{1}{2}\exp\left(-\log\left[2(1 - p_0)\right] - \frac{r}{b}\right)$.

iii) $b \log \lambda \leq -r \Leftrightarrow \lambda \leq \exp\left(-\frac{r}{b}\right)$
the bound is
$$\max_{\lambda \leq \exp\left(-\frac{r}{b}\right)} \lambda \cdot p_0 = p_0 \exp\left(-\frac{r}{b}\right).$$

Since $\hat{\lambda} > e^{r/b} \Leftrightarrow p_0 > 1 - \frac{1}{2}\exp(-\frac{r}{b})$, notice that the lower bound is a concave function w.r.t. $\lambda$, making the final lower bound become
$$\begin{cases} 1 - \exp\left(\frac{r}{b}\right)(1 - p_0), & \text{when} \quad p_0 > 1 - \frac{1}{2}\exp(-\frac{r}{b}) \\ \frac{1}{2}\exp\left(-\log\left[2(1 - p_0)\right] - \frac{r}{b}\right). & \text{otherwise} \end{cases}$$

$\square$

**Remark** Actually, we have $1 - \exp\left(\frac{r}{b}\right)(1 - p_0) \leq \frac{1}{2}\exp\left(-\log\left[2(1 - p_0)\right] - \frac{r}{b}\right)$ all the time. Another interesting thing is that both the bound can lead to the same radius bound:
$$1 - \exp\left(\frac{r}{b}\right)(1 - p_0) > \frac{1}{2} \Leftrightarrow r < -b \log\left[2(1 - p_0)\right]$$
$$\frac{1}{2}\exp\left(-\log\left[2(1 - p_0)\right] - \frac{r}{b}\right) > \frac{1}{2} \Leftrightarrow r < -b \log\left[2(1 - p_0)\right]$$

### A.3 Proof for Corollary 2

*Proof.* With strong duality, our confidence lower bound is
$$\min_{\|\boldsymbol{\delta}\|_2 \leq r} \max_{\lambda \geq 0} \left\{\lambda p_0 - \int (\lambda \pi_{\mathbf{0}}(z) - \pi_{\boldsymbol{\delta}}(z))_+ \, dz\right\},$$

define $C_\lambda = \{z : \lambda \pi_{\mathbf{0}}(z) \geq \pi_{\boldsymbol{\delta}}(z)\} = \{z : \boldsymbol{\delta}^\top z \leq \frac{\|\boldsymbol{\delta}\|^2}{2} + \sigma^2 \ln \lambda\}$ and $\Phi(\cdot)$ to be the cdf of standard gaussian distribution, then
$$\int (\lambda \pi_{\mathbf{0}}(z) - \pi_{\boldsymbol{\delta}}(z))_+ \, dz$$
$$= \int_{C_\lambda} (\lambda \pi_{\mathbf{0}}(z) - \pi_{\boldsymbol{\delta}}(z)) \, dz$$
$$= \lambda \cdot \mathbb{P}\left(N(z; \mathbf{0}, \sigma^2 \boldsymbol{I}) \in C_\lambda\right) - \mathbb{P}\left(N(z; \boldsymbol{\delta}, \sigma^2 \boldsymbol{I}) \in C_\lambda\right)$$
$$= \lambda \cdot \Phi\left(\frac{\|\boldsymbol{\delta}\|_2}{2\sigma} + \frac{\sigma \ln \lambda}{\|\boldsymbol{\delta}\|_2}\right) - \Phi\left(\frac{-\|\boldsymbol{\delta}\|_2}{2\sigma} + \frac{\sigma \ln \lambda}{\|\boldsymbol{\delta}\|_2}\right).$$

Define

$$F(\boldsymbol{\delta}, \lambda) := \lambda p_0 - \int \left(\lambda \pi_{\mathbf{0}}(z) - \pi_{\boldsymbol{\delta}}(z)\right)_+ dz = \lambda p_0 - \lambda \cdot \Phi\left(\frac{\|\boldsymbol{\delta}\|_2}{2\sigma} + \frac{\sigma \ln \lambda}{\|\boldsymbol{\delta}\|_2}\right) + \Phi\left(\frac{-\|\boldsymbol{\delta}\|_2}{2\sigma} + \frac{\sigma \ln \lambda}{\|\boldsymbol{\delta}\|_2}\right).$$

For $\forall \boldsymbol{\delta}$, $F$ is a concave function w.r.t. $\lambda$, as $F$ is actually a summation of many concave piece wise linear function. See [33] for more discussions of properties of concave functions.

Define $\hat{\lambda}_{\boldsymbol{\delta}} = \exp\left(\frac{2\sigma\|\boldsymbol{\delta}\|_2 \Phi^{-1}(p_0) - \|\boldsymbol{\delta}\|_2^2}{2\sigma^2}\right)$, simple calculation can show $\frac{\partial F(\boldsymbol{\delta}, \lambda)}{\partial \lambda}|_{\lambda = \hat{\lambda}_{\boldsymbol{\delta}}} = 0$, which means

$$\min_{\|\boldsymbol{\delta}\|_2 \le r} \max_{\lambda \ge 0} F(\boldsymbol{\delta}, \lambda) = \min_{\|\boldsymbol{\delta}\|_2 \le r} F(\boldsymbol{\delta}, \lambda_{\boldsymbol{\delta}})$$

$$= \min_{\|\boldsymbol{\delta}\|_2 \le r} \left\{ 0 + \Phi\left(\frac{-\|\boldsymbol{\delta}\|_2}{2\sigma} + \frac{\sigma \ln \hat{\lambda}_{\boldsymbol{\delta}}}{\|\boldsymbol{\delta}\|_2}\right) \right\}$$

$$= \min_{\|\boldsymbol{\delta}\|_2 \le r} \Phi\left(\Phi^{-1}(p_0) - \frac{\|\boldsymbol{\delta}\|_2}{\sigma}\right)$$

$$= \Phi\left(\Phi^{-1}(p_0) - \frac{r}{\sigma}\right)$$

This tells us

$$\min_{\|\boldsymbol{\delta}\|_2 \le r} \max_{\lambda \ge 0} F(\boldsymbol{\delta}, \lambda) > 1/2 \Leftrightarrow \Phi\left(\Phi^{-1}(p_0) - \frac{r}{\sigma}\right) > 1/2 \Leftrightarrow r < \sigma \cdot \Phi^{-1}(p_0),$$

i.e. the certification radius is $\sigma \cdot \Phi^{-1}(p_0)$. This is exactly the core theoretical contribution of [9]. This bound has a straight forward expansion for multi-class classification situations, we refer interesting readers to Appendix C. ☐

## A.4 Proof For Theorem 2 and 3

### A.4.1 Proof for $\ell_2$ and $\ell_\infty$ cases

Here we consider a more general smooth distribution $\pi_{\mathbf{0}}(\boldsymbol{z}) \propto \|\boldsymbol{z}\|_\infty^{-k_1} \|\boldsymbol{z}\|_2^{-k_2} \exp\left(-\frac{\|\boldsymbol{z}\|_2^2}{2\sigma^2}\right)$, for some $k_1, k_2 \ge 0$ and $\sigma > 0$. We first gives the following key theorem shows that $\mathbb{D}_{\mathcal{F}_{[0,1]}}\left(\lambda \pi_{\mathbf{0}} \| \pi_{\boldsymbol{\delta}}\right)$ increases as $|\delta_i|$ becomes larger for every dimension $i$.

**Theorem 4.** Suppose $\pi_{\mathbf{0}}(\boldsymbol{z}) \propto \|\boldsymbol{z}\|_\infty^{-k_1} \|\boldsymbol{z}\|_2^{-k_2} \exp\left(-\frac{\|\boldsymbol{z}\|_2^2}{2\sigma^2}\right)$, for some $k_1, k_2 \ge 0$ and $\sigma > 0$, for any $\lambda \ge 0$ we have

$$\operatorname{sgn}(\delta_i) \frac{\partial}{\partial \delta_i} \mathbb{D}_{\mathcal{F}_{[0,1]}}\left(\lambda \pi_{\mathbf{0}} \| \pi_{\boldsymbol{\delta}}\right) \ge 0,$$

for any $i \in \{1, 2, ..., d\}$.

Theorem 2 and 3 directly follows the above theorem. Notice that in Theorem 2, as our distribution is spherical symmetry, it is equivalent to set $\mathcal{B} = \left\{\boldsymbol{\delta} : \boldsymbol{\delta} = [a, 0, ..., 0]^\top, a \le r\right\}$ by rotating the axis.

*Proof.* Given $\lambda$, $k_1$ and $k_2$, we define $\phi_1(s) = s^{-k_1}$, $\phi_2(s) = s^{-k_2} e^{-\frac{s^2}{\sigma^2}}$. Notice that $\phi_1$ and $\phi_2$ are monotone decreasing for non-negative $s$. By the symmetry, without loss of generality, we assume $\boldsymbol{\delta} = [\delta_1, ..., \delta_d]^\top$ for $\delta_i \ge 0$, $i \in [d]$. Notice that

$$\frac{\partial}{\partial \delta_i} \|\boldsymbol{x}_0 - \boldsymbol{\delta}\|_\infty = \mathbb{I}\{\|\boldsymbol{x}_0 - \boldsymbol{\delta}\|_\infty = |x_i - \delta_i|\} \frac{\partial}{\partial \delta_i} \sqrt{(x_i - \delta_i)^2}$$

$$= \mathbb{I}\{\|\boldsymbol{x}_0 - \boldsymbol{\delta}\|_\infty = |x_i - \delta_i|\} \frac{-(x_i - \delta_i)}{\|\boldsymbol{x}_0 - \boldsymbol{\delta}\|_\infty}.$$

And also

$$\frac{\partial}{\partial \delta_i} \|\boldsymbol{x}_0 - \boldsymbol{\mu}\|_2 = \frac{\partial}{\partial \delta_i} \sqrt{\sum_i (x_i - \mu_i)^2}$$

$$= \frac{-(x_i - \mu_i)}{\|\boldsymbol{x}_0 - \boldsymbol{\mu}\|_2}.$$

We thus have

$$\frac{\partial}{\partial \delta_1} \int \left( \lambda \pi_{\boldsymbol{0}}(\boldsymbol{x}_0) - \pi_{\boldsymbol{\delta}}(\boldsymbol{x}_0) \right)_+ d\boldsymbol{x}_0$$

$$= -\int \mathbb{I}\{\lambda \pi_{\boldsymbol{0}}(\boldsymbol{x}_0) \geq \pi_{\boldsymbol{\delta}}(\boldsymbol{x}_0)\} \frac{\partial}{\partial \delta_1} \pi_{\boldsymbol{\delta}}(\boldsymbol{x}_0) d\boldsymbol{x}_0$$

$$= \int \mathbb{I}\{\lambda \pi_{\boldsymbol{0}}(\boldsymbol{x}_0) \geq \pi_{\boldsymbol{\delta}}(\boldsymbol{x}_0)\} F_1 \left( \|\boldsymbol{x}_0 - \boldsymbol{\delta}\|_\infty, \|\boldsymbol{x}_0 - \boldsymbol{\delta}\|_2 \right) d\boldsymbol{x}_0$$

$$= \int \mathbb{I}\{\lambda \pi_{\boldsymbol{0}}(\boldsymbol{x}_0) \geq \pi_{\boldsymbol{\delta}}(\boldsymbol{x}_0), x_1 > \delta_1\} F_1 \left( \|\boldsymbol{x}_0 - \boldsymbol{\delta}\|_\infty, \|\boldsymbol{x}_0 - \boldsymbol{\delta}\|_2 \right) d\boldsymbol{x}_0$$

$$+ \int \mathbb{I}\{\lambda \pi_{\boldsymbol{0}}(\boldsymbol{x}_0) \geq \pi_{\boldsymbol{\delta}}(\boldsymbol{x}_0), x_1 < \delta_1\} F_1 \left( \|\boldsymbol{x}_0 - \boldsymbol{\delta}\|_\infty, \|\boldsymbol{x}_0 - \boldsymbol{\delta}\|_2 \right) d\boldsymbol{x}_0,$$

where we define

$$F_1 \left( \|\boldsymbol{x}_0 - \boldsymbol{\delta}\|_\infty, \|\boldsymbol{x}_0 - \boldsymbol{\delta}\|_2 \right)$$

$$= \phi_1' \left( \|\boldsymbol{x}_0 - \boldsymbol{\delta}\|_\infty \right) \phi_2 \left( \|\boldsymbol{x}_0 - \boldsymbol{\delta}\|_2 \right) \mathbb{I}\{\|\boldsymbol{x}_0 - \boldsymbol{\delta}\|_\infty = |x_1 - \delta_1|\} \frac{(x_1 - \delta_1)}{\|\boldsymbol{x}_0 - \boldsymbol{\delta}\|_\infty}$$

$$+ \phi_1 \left( \|\boldsymbol{x}_0 - \boldsymbol{\delta}\|_\infty \right) \phi_2' \left( \|\boldsymbol{x}_0 - \boldsymbol{\delta}\|_2 \right) \frac{(x_1 - \delta_1)}{\|\boldsymbol{x}_0 - \boldsymbol{\delta}\|_2}.$$

Notice that as $\phi_1' \leq 0$ and $\phi_2' \leq 0$ and we have

$$\int \mathbb{I}\{\lambda \pi_{\boldsymbol{0}}(\boldsymbol{x}_0) \geq \pi_{\boldsymbol{\delta}}(\boldsymbol{x}_0), x_1 > \delta_1\} F_1 \left( \|\boldsymbol{x}_0 - \boldsymbol{\delta}\|_\infty, \|\boldsymbol{x}_0 - \boldsymbol{\delta}\|_2 \right) d\boldsymbol{x}_0 \leq 0$$

$$\int \mathbb{I}\{\lambda \pi_{\boldsymbol{0}}(\boldsymbol{x}_0) \geq \pi_{\boldsymbol{\delta}}(\boldsymbol{x}_0), x_1 < \delta_1\} F_1 \left( \|\boldsymbol{x}_0 - \boldsymbol{\delta}\|_\infty, \|\boldsymbol{x}_0 - \boldsymbol{\delta}\|_2 \right) d\boldsymbol{x}_0 \geq 0.$$

Our target is to prove that $\frac{\partial}{\partial \delta_1} \int \left( \lambda \pi_{\boldsymbol{0}}(\boldsymbol{x}_0) - \pi_{\boldsymbol{\delta}}(\boldsymbol{x}_0) \right)_+ d\boldsymbol{x}_0 \geq 0$. Now define the set

$$H_1 = \{\boldsymbol{x}_0 : \lambda \pi_{\boldsymbol{0}}(\boldsymbol{x}_0) \geq \pi_{\boldsymbol{\delta}}(\boldsymbol{x}_0), x_1 > \delta_1\}$$

$$H_2 = \left\{ [2\delta_1 - x_1, x_2, ..., x_d]^\top : \boldsymbol{x}_0 = [x_1, ..., x_d]^\top \in H_1 \right\}.$$

Here the set $H_2$ is defined as a image of a bijection

$$\text{proj}(\boldsymbol{x}_0) = [2\delta_1 - x_1, x_2, ..., x_d]^\top = \tilde{\boldsymbol{x}}_0,$$

that is constrained on the set $H_1$. Notice that under our definition,

$$\int \mathbb{I}\{\lambda \pi_{\boldsymbol{0}}(\boldsymbol{x}_0) \geq \pi_{\boldsymbol{\delta}}(\boldsymbol{x}_0), x_1 > \delta_1\} F_1 \left( \|\boldsymbol{x}_0 - \boldsymbol{\delta}\|_\infty, \|\boldsymbol{x}_0 - \boldsymbol{\delta}\|_2 \right) d\boldsymbol{x}_0$$

$$= \int_{H_1} F_1 \left( \|\boldsymbol{x}_0 - \boldsymbol{\delta}\|_\infty, \|\boldsymbol{x}_0 - \boldsymbol{\delta}\|_2 \right) d\boldsymbol{x}_0.$$

Now we prove that

$$\int \mathbb{I}\{\lambda \pi_{\boldsymbol{0}}(\boldsymbol{x}_0) \geq \pi_{\boldsymbol{\delta}}(\boldsymbol{x}_0), x_1 < \delta_1\} F_1 \left( \|\boldsymbol{x}_0 - \boldsymbol{\delta}\|_\infty, \|\boldsymbol{x}_0 - \boldsymbol{\delta}\|_2 \right) d\boldsymbol{x}_0$$

$$\overset{(1)}{\geq} \int_{H_2} F_1 \left( \|\boldsymbol{x}_0 - \boldsymbol{\delta}\|_\infty, \|\boldsymbol{x}_0 - \boldsymbol{\delta}\|_2 \right) d\boldsymbol{x}_0$$

$$\overset{(2)}{=} \left| \int_{H_1} F_1 \left( \|\boldsymbol{x}_0 - \boldsymbol{\delta}\|_\infty, \|\boldsymbol{x}_0 - \boldsymbol{\delta}\|_2 \right) d\boldsymbol{x}_0 \right|.$$

**Property of the projection** Before we prove the (1) and (2), we give the following property of the defined projection function. For any $\tilde{\boldsymbol{x}}_0 = \mathrm{proj}(\boldsymbol{x}_0)$, $\boldsymbol{x}_0 \in H_1$, we have

$$\|\boldsymbol{x}_0 - \boldsymbol{\delta}\|_\infty = \|\tilde{\boldsymbol{x}}_0 - \boldsymbol{\delta}\|_\infty$$
$$\|\boldsymbol{x}_0 - \boldsymbol{\delta}\|_2 = \|\tilde{\boldsymbol{x}}_0 - \boldsymbol{\delta}\|_2$$
$$\|\boldsymbol{x}_0\|_2 \geq \|\tilde{\boldsymbol{x}}_0\|_2$$
$$\|\boldsymbol{x}_0\|_\infty \geq \|\tilde{\boldsymbol{x}}_0\|_\infty \, .$$

This is because

$$\tilde{x}_i = x_i, i \in [d] - \{1\}$$
$$\tilde{x}_1 = 2\delta_1 - x_1,$$

and by the fact that $x_1 \geq \delta_1 \geq 0$, we have $|\tilde{x}_1| \leq |x_1|$ and $|\tilde{x}_1 - \delta_1| \leq |x_1 - \delta_1|$.

**Proof of Equality (2)** By the fact that $\mathrm{proj}$ is bijective constrained on the set $H_1$ and the property of $\mathrm{proj}$, we have

$$\int_{H_2} F_1 \left( \|\tilde{\boldsymbol{x}}_0 - \boldsymbol{\delta}\|_\infty, \|\tilde{\boldsymbol{x}}_0 - \boldsymbol{\delta}\|_2 \right) d\tilde{\boldsymbol{x}}_0$$

$$= \int_{H_2} \phi_1' \left( \|\tilde{\boldsymbol{x}}_0 - \boldsymbol{\delta}\|_\infty \right) \phi_2 \left( \|\tilde{\boldsymbol{x}}_0 - \boldsymbol{\delta}\|_2 \right) \mathbb{I}\{ \|\tilde{\boldsymbol{x}}_0 - \boldsymbol{\delta}\|_\infty = |\tilde{x}_1 - \delta_1| \} \frac{(\tilde{x}_1 - \delta_1)}{\|\tilde{\boldsymbol{x}}_0 - \boldsymbol{\delta}\|_\infty} d\tilde{\boldsymbol{x}}_0$$

$$+ \int_{H_2} \phi_1 \left( \|\tilde{\boldsymbol{x}}_0 - \boldsymbol{\delta}\|_\infty \right) \phi_2' \left( \|\tilde{\boldsymbol{x}}_0 - \boldsymbol{\delta}\|_2 \right) \frac{(\tilde{x}_1 - \delta_1)}{\|\tilde{\boldsymbol{x}}_0 - \boldsymbol{\delta}\|_2} d\tilde{\boldsymbol{x}}_0$$

$$\stackrel{(*)}{=} \int_{H_1} \phi_1' \left( \|\boldsymbol{x}_0 - \boldsymbol{\delta}\|_\infty \right) \phi_2 \left( \|\boldsymbol{x}_0 - \boldsymbol{\delta}\|_2 \right) \mathbb{I}\{ \|\boldsymbol{x}_0 - \boldsymbol{\delta}\|_\infty = |x_1 - \delta_1| \} \frac{(\delta_1 - x_1)}{\|\boldsymbol{x}_0 - \boldsymbol{\delta}\|_\infty} |\det(\boldsymbol{J})| \, d\boldsymbol{x}_0$$

$$+ \int_{H_1} \phi_1 \left( \|\boldsymbol{x}_0 - \boldsymbol{\delta}\|_\infty \right) \phi_2' \left( \|\boldsymbol{x}_0 - \boldsymbol{\delta}\|_2 \right) \frac{(\delta_1 - x_1)}{\|\boldsymbol{x}_0 - \boldsymbol{\delta}\|_2} d\boldsymbol{x}_0$$

$$= - \int_{H_1} F_1 \left( \|\boldsymbol{x}_0 - \boldsymbol{\delta}\|_\infty, \|\boldsymbol{x}_0 - \boldsymbol{\delta}\|_2 \right) d\boldsymbol{x}_0,$$

where $(*)$ is by change of variable $\tilde{\boldsymbol{x}}_0 = \mathrm{proj}(\boldsymbol{x}_0)$ and $\boldsymbol{J}$ is the Jacobian matrix $\boldsymbol{J} = \begin{bmatrix} -1 & 0 & \cdots & 0 \\ 0 & 1 & \cdots & 0 \\ \vdots & \vdots & \ddots & \vdots \\ 0 & 0 & \cdots & 1 \end{bmatrix}$ and here we have the fact that $\tilde{x}_1 - \delta_1 = (2\delta_1 - x_1) - \delta_1 = -(x_1 - \delta_1)$.

**Proof of Inequality (1)** This can be done by verifying that $H_2 \subseteq \{\boldsymbol{x}_0 : \lambda\pi_{\boldsymbol{0}}(\boldsymbol{x}_0) \geq \pi_{\boldsymbol{\delta}}(\boldsymbol{x}_0), x_1 < \delta_1\}$. By the property of the projection, for any $\boldsymbol{x}_0 \in H_1$, let $\tilde{\boldsymbol{x}}_0 = \mathrm{proj}(\boldsymbol{x}_0)$, then $\lambda\pi_{\boldsymbol{0}}(\tilde{\boldsymbol{x}}_0) \geq \lambda\pi_{\boldsymbol{0}}(\boldsymbol{x}_0) \geq \pi_{\boldsymbol{\delta}}(\boldsymbol{x}_0) = \pi_{\boldsymbol{\delta}}(\tilde{\boldsymbol{x}}_0)$ (by the fact that t $\phi_1$ and $\phi_2$ are monotone decreasing). It implies that for any $\tilde{\boldsymbol{x}}_0 \in H_2$, we have $\lambda\pi_{\boldsymbol{0}}(\tilde{\boldsymbol{x}}_0) \geq \pi_{\boldsymbol{\delta}}(\tilde{\boldsymbol{x}}_0)$ and thus $H_2 \subseteq \{\boldsymbol{x}_0 : \pi_{\boldsymbol{0}}(\boldsymbol{x}_0) \geq \pi_{\boldsymbol{\delta}}(\boldsymbol{x}_0), x_1 < \delta_1\}$.

**Final statement** By the above result, we have

$$\frac{\partial}{\partial \delta_1} \int \left( \lambda\pi_{\boldsymbol{0}}(\boldsymbol{x}_0) - \pi_{\boldsymbol{\delta}}(\boldsymbol{x}_0) \right)_+ d\boldsymbol{x}_0 \geq 0,$$

and the same result holds for any $\frac{\partial}{\partial \delta_1} \int \left( \lambda\pi_{\boldsymbol{0}}(\boldsymbol{x}_0) - \pi_{\boldsymbol{\delta}}(\boldsymbol{x}_0) \right)_+ d\boldsymbol{x}_0, i \in [d]$, which implies our result. $\qquad \square$

### A.4.2 Proof for $\ell_1$ case

Slightly different for former cases, apart from proving $\frac{\partial}{\partial \delta_i} \mathbb{D}_{\mathcal{F}_{[0,1]}} \left( \lambda\pi_{\boldsymbol{0}} \| \pi_{\boldsymbol{\delta}} \right) \geq 0$ for $\forall \delta_i \geq 0$, we also need to demonstrate

**Theorem 5.** *Suppose* $\pi_{\boldsymbol{0}}(\boldsymbol{x}_0) \propto \|\boldsymbol{x}_0\|^{-k} \exp \left( -\frac{\|\boldsymbol{x}_0\|_1}{b} \right)$, *then for* $\boldsymbol{\delta} = (r, d - r, \delta_3, \delta_4, \cdots)$ *and* $\tilde{\boldsymbol{\delta}} = (0, d, \delta_3, \delta_4, \cdots)$, $0 < r < d$, *we have*

$$\mathbb{D}_{\mathcal{F}_{[0,1]}} \left( \lambda\pi_{\boldsymbol{0}} \| \pi_{\boldsymbol{\delta}} \right) \geq \mathbb{D}_{\mathcal{F}_{[0,1]}} \left( \lambda\pi_{\boldsymbol{0}} \| \pi_{\tilde{\boldsymbol{\delta}}} \right)$$

*Proof.* We turn to show that

$$\frac{\partial}{\partial r}\mathbb{D}_{\mathcal{F}_{[0,1]}}\left(\lambda\pi_{\mathbf{0}}\parallel\pi_{\boldsymbol{\delta}}\right)\leq 0,$$

for $\boldsymbol{\delta}=(r,d-r,\delta_3,\delta_4,\cdots)$ and $r<d/2$. We define $\phi(s)=s^{-k}\exp(-\frac{s}{b})$. With

$$\frac{\partial}{\partial\delta_i}\|\boldsymbol{x}_0-\boldsymbol{\delta}\|_1=\frac{\partial}{\partial\delta_i}|x_i-\delta_i|=-\mathrm{sgn}(x_i-\delta_i)=\frac{\delta_i-x_i}{|x_i-\delta_i|},$$

We have

$$\frac{\partial}{\partial r}\mathbb{D}_{\mathcal{F}_{[0,1]}}\left(\lambda\pi_{\mathbf{0}}\parallel\pi_{\boldsymbol{\delta}}\right)$$

$$=-\int\mathbb{I}\{\lambda\pi_{\mathbf{0}}(\boldsymbol{x}_0)\geq\pi_{\boldsymbol{\delta}}(\boldsymbol{x}_0)\}\frac{\partial}{\partial r}\pi_{\boldsymbol{\delta}}(\boldsymbol{x}_0)d\boldsymbol{x}_0$$

$$=\int\mathbb{I}\{\lambda\pi_{\mathbf{0}}(\boldsymbol{x}_0)\geq\pi_{\boldsymbol{\delta}}(\boldsymbol{x}_0)\}F(\boldsymbol{x}_0)d\boldsymbol{x}_0,$$

where

$$F(\boldsymbol{x}_0)=-\frac{\partial}{\partial r}\phi\left(\|\boldsymbol{x}_0-\boldsymbol{\delta}\|_1\right)=-\phi'\left(\|\boldsymbol{x}_0-\boldsymbol{\delta}\|_1\right)\frac{\partial}{\partial r}\|\boldsymbol{x}_0-\boldsymbol{\delta}\|_1$$

$$=\phi'\left(\|\boldsymbol{x}_0-\boldsymbol{\delta}\|_1\right)\frac{\partial}{\partial r}\left(|x_1-r|+|x_2-d+r|\right)$$

$$=\phi'\left(\|\boldsymbol{x}_0-\boldsymbol{\delta}\|_1\right)\cdot\left(\mathrm{sgn}(x_1-r)+\mathrm{sgn}(d-x_2-r)\right).$$

Thus the original derivative becomes

$$=\int\mathbb{I}\{\lambda\pi_{\mathbf{0}}(\boldsymbol{x}_0)\geq\pi_{\boldsymbol{\delta}}(\boldsymbol{x}_0),x_1>r,x_2<d-r\}F(\boldsymbol{x}_0)d\boldsymbol{x}_0$$

$$+\int\mathbb{I}\{\lambda\pi_{\mathbf{0}}(\boldsymbol{x}_0)\geq\pi_{\boldsymbol{\delta}}(\boldsymbol{x}_0),x_1>r,x_2>d-r\}F(\boldsymbol{x}_0)d\boldsymbol{x}_0$$

$$+\int\mathbb{I}\{\lambda\pi_{\mathbf{0}}(\boldsymbol{x}_0)\geq\pi_{\boldsymbol{\delta}}(\boldsymbol{x}_0),x_1<r,x_2>d-r\}F(\boldsymbol{x}_0)d\boldsymbol{x}_0$$

$$+\int\mathbb{I}\{\lambda\pi_{\mathbf{0}}(\boldsymbol{x}_0)\geq\pi_{\boldsymbol{\delta}}(\boldsymbol{x}_0),x_1<r,x_2<d-r\}F(\boldsymbol{x}_0)d\boldsymbol{x}_0$$

$$=2\int\mathbb{I}\{\lambda\pi_{\mathbf{0}}(\boldsymbol{x}_0)\geq\pi_{\boldsymbol{\delta}}(\boldsymbol{x}_0),x_1>r,x_2<d-r\}\phi'(\|\boldsymbol{x}_0-\boldsymbol{\delta}\|_1)d\boldsymbol{x}_0$$

$$-2\int\mathbb{I}\{\lambda\pi_{\mathbf{0}}(\boldsymbol{x}_0)\geq\pi_{\boldsymbol{\delta}}(\boldsymbol{x}_0),x_1<r,x_2>d-r\}\phi'(\|\boldsymbol{x}_0-\boldsymbol{\delta}\|_1)d\boldsymbol{x}_0$$

We only need to show that

$$\int\mathbb{I}\{\lambda\pi_{\mathbf{0}}(\boldsymbol{x}_0)\geq\pi_{\boldsymbol{\delta}}(\boldsymbol{x}_0),x_1>r,x_2<d-r\}\phi'(\|\boldsymbol{x}_0-\boldsymbol{\delta}\|_1)d\boldsymbol{x}_0\geq$$

$$\int\mathbb{I}\{\lambda\pi_{\mathbf{0}}(\boldsymbol{x}_0)\geq\pi_{\boldsymbol{\delta}}(\boldsymbol{x}_0),x_1<r,x_2>d-r\}\phi'(\|\boldsymbol{x}_0-\boldsymbol{\delta}\|_1)d\boldsymbol{x}_0.$$

Notice that $r<d/2$, therefore this can be proved with a similar projection $\boldsymbol{x}_0\mapsto\tilde{\boldsymbol{x}}_0$:

$$(x_1,x_2,x_3,x_4,\cdots)\mapsto(2r-x_1,2d-2r-x_2,x_3,x_4,\cdots)$$

and the similar deduction as previous theorem.

$$\square$$

## A.5   Theoretical Demonstration about the Ineffetivity of Equation (12)

**Theorem 6.** *Consider the adversarial attacks on the $\ell_\infty$ ball $\mathcal{B}_{\ell_\infty,r}=\{\boldsymbol{\delta}:\|\boldsymbol{\delta}\|_\infty\leq r\}$. Suppose we use the smoothing distribution $\pi_{\mathbf{0}}$ in Equation (12) and choose the parameters $(k,\sigma)$ such that*

*1) $\|z\|_\infty$ is stochastic bounded when $z \sim \pi_0$, in that for any $\epsilon > 0$, there exists a finite $M > 0$ such that $P_{\pi_0}(|z| > M) \le \epsilon$;*

*2) the mode of $\|z\|_\infty$ under $\pi_0$ equals $Cr$, where $C$ is some fixed positive constant,*

*then for any $\epsilon \in (0,1)$ and sufficiently large dimension $d$, there exists a constant $t > 1$, such that , we have*

$$\max_{\delta \in \mathcal{B}_{\ell_\infty, r}} \left\{ \mathbb{D}_{\mathcal{F}_{[0,1]}} (\lambda \pi_0 \| \pi_\delta) \right\} \ge (1 - \epsilon)(\lambda - \mathcal{O}(t^{-d})).$$

*This shows that, in very high dimensions, the maximum distance term is arbitrarily close to $\lambda$ which is the maximum possible value of $\mathbb{D}_{\mathcal{F}_{[0,1]}} (\lambda \pi_0 \| \pi_\delta)$ (see Theorem 1). In particular, this implies that in high dimensional scenario, once $f_{\pi_0}^\sharp(x_0) \le (1-\epsilon)$ for some small $\epsilon$, we have $\mathcal{L}_{\pi_0}(\mathcal{F}_{[0,1]}, \mathcal{B}_{\ell_\infty, r}) = \mathcal{O}(t^{-d})$ and thus fail to certify.*

**Remark** The condition 1) and 2) in Theorem 6 are used to ensure that the magnitude of the random perturbations generated by $\pi_0$ is within a reasonable range such that the value of $f_{\pi_0}^\sharp(x_0)$ is not too small, in order to have a high accuracy in the trade-off in Equation (9). Note that the natural images are often contained in cube $[0,1]^d$. If $\|z\|_\infty$ is too large to exceed the region of natural images, the accuracy will be obviously rather poor. Note that if we use variants of Gaussian distribution, we only need $\|z\|_2/\sqrt{d}$ to be not too large. Theorem 6 says that once $\|z\|_\infty$ is in a reasonably small scale, the maximum distance term must be unreasonably large in high dimensions, yielding a vacuous lower bound.

*Proof.* First notice that the distribution of $z$ can be factorized by the following hierarchical scheme:

$$a \sim \pi_R(a) \propto a^{d-1-k} e^{-\frac{a^2}{2\sigma^2}} \mathbb{I}\{a \ge 0\}$$
$$s \sim \text{Unif}^{\otimes d}(-1, 1)$$
$$z \leftarrow \frac{s}{\|s\|_\infty} a.$$

Without loss of generality, we assume $\delta^* = [r, ..., r]^\top$. (see Theorem 4)

$$\mathbb{D}_{\mathcal{F}_{[0,1]}} (\lambda \pi_0 \| \pi_{\delta^*}) = \mathbb{E}_{z \sim \pi_0} \left( \lambda - \frac{\pi_\delta}{\pi_0}(z) \right)_+.$$

Notice that as the distribution is symmetry,

$$P_{\pi_0} (\|z + \delta^*\|_\infty = a + r \mid \|z\|_\infty = a) = \frac{1}{2}.$$

Define $|z|^{(i)}$ is the $i$-th order statistics of $|z_j|$, $j = 1, ..., d$ conditioning on $\|z\|_\infty = a$. By the factorization above and some algebra, we have, for any $\epsilon \in (0,1)$,

$$P \left( \frac{|z|^{(d-1)}}{|z|^{(d)}} > (1 - \epsilon) \mid \|z\|_\infty = a \right) \ge 1 - (1 - \epsilon)^{d-1}.$$

And $\frac{|z|^{(d-1)}}{|z|^{(d)}} \perp |z|^{(d)}$. Now we estimate $\mathbb{D}_{\mathcal{F}_{[0,1]}} (\lambda \pi_0 \| \pi_{\delta^*})$.

$$\mathbb{E}_{z \sim \pi_0} \left( \lambda - \frac{\pi_\delta}{\pi_0}(z) \right)_+$$
$$= \mathbb{E}_a \mathbb{E}_{z \sim \pi_0} \left[ \left( \lambda - \frac{\pi_\delta}{\pi_0}(z) \right)_+ \mid \|z\|_\infty = a \right]$$
$$= \frac{1}{2} \mathbb{E}_a \mathbb{E}_{z \sim \pi_0} \left[ \left( \lambda - \frac{\pi_\delta}{\pi_0}(z) \right)_+ \mid \|z\|_\infty = a, \|z + \delta^*\|_\infty = a + r \right]$$
$$+ \frac{1}{2} \mathbb{E}_a \mathbb{E}_{z \sim \pi_0} \left[ \left( \lambda - \frac{\pi_\delta}{\pi_0}(z) \right)_+ \mid \|z\|_\infty = a, \|z + \delta^*\|_\infty \ne a + r \right].$$

Conditioning on $\|z\|_\infty = a, \|z + \delta^*\|_\infty = a + r$, we have

$$\frac{\pi_\delta}{\pi_0}(z) = \left(\frac{1}{1 + \frac{r}{a}}\right)^k e^{-\frac{1}{2\sigma^2}\left(2ra + r^2\right)}$$

$$= \left(\frac{1}{1 + \frac{r}{a}}\right)^k e^{-\frac{d-1-k}{2C^2}\left(2\frac{a}{r} + 1\right)}.$$

Here the second equality is because we choose $\mathrm{mode}(\|z\|_\infty) = Cr$, which implies that $\sqrt{d - 1 - k}\,\sigma = Cr$. And thus we have

$$\mathbb{E}_a \mathbb{E}_{z \sim \pi_0}\left[\left(\lambda - \frac{\pi_\delta}{\pi_0}(z)\right)_+ \mid \|z\|_\infty = a, \|z + \delta^*\|_\infty = a + r\right]$$

$$= \int \left(\lambda - \left(\frac{1}{1 + \frac{r}{a}}\right)^k e^{-\frac{d-1-k}{2C^2}\left(2\frac{a}{r} + 1\right)}\right)_+ \pi(a)\,da$$

$$= \int \left(\lambda - \left(1 + \frac{r}{a}\right)^{-k} \left(e^{\frac{2a/r + 1}{2C^2}}\right)^{-(d-1-k)}\right)_+ \pi(a)\,da$$

$$= \lambda - \mathcal{O}(t^{-d}),$$

for some $t > 1$. Here the last equality is by the assumption that $\|z\|_\infty = \mathcal{O}_p(1)$.

Next we bound the second term $\mathbb{E}_a \mathbb{E}_{z \sim \pi_0}\left[\left(\lambda - \frac{\pi_\delta}{\pi_0}(z)\right)_+ \mid \|z\|_\infty = a, \|z + \delta^*\|_\infty \neq a + r\right]$. By the property of uniform distribution, we have

$$\mathrm{P}\left(\frac{|z|^{(d-1)}}{|z|^{(d)}} > (1 - \epsilon) \mid \|z\|_\infty = a, \|z + \delta^*\|_\infty \neq a + r\right)$$

$$= \mathrm{P}\left(\frac{|z|^{(d-1)}}{|z|^{(d)}} > (1 - \epsilon) \mid \|z\|_\infty = a\right)$$

$$\geq 1 - (1 - \epsilon)^{d-1}.$$

And thus, for any $\epsilon \in [0, 1)$,

$$\mathrm{P}\left(\|z + \delta^*\|_\infty \geq ((1 - \epsilon)a + r)^2 \mid \|z\|_\infty = a, \|z + \delta^*\|_\infty \neq a + r\right) \geq \frac{1}{2}\left(1 - (1 - \epsilon)^{d-1}\right).$$

It implies that

$$\mathbb{E}_{z \sim \pi_0}\left[\left(\lambda - \frac{\pi_\delta}{\pi_0}(z)\right)_+ \mid \|z\|_\infty = a, \|z + \delta^*\|_\infty = a + r\right]$$

$$\geq \frac{1}{2}\left(1 - (1 - \epsilon)^{d-1}\right)\left(\lambda - \left(1 - \epsilon + \frac{r}{a}\right)^{-k} e^{-\frac{1}{2\sigma^2}\left(\epsilon(\epsilon - 2)a^2 + 2r(1 - \epsilon)a + r^2\right)}\right)_+$$

$$= \frac{1}{2}\left(1 - (1 - \epsilon)^{d-1}\right)\left(\lambda - \left(1 - \epsilon + \frac{r}{a}\right)^{-k} e^{-\frac{d-1-k}{2C^2}\left(\epsilon(\epsilon - 2)a^2/r^2 + 2(1 - \epsilon)a/r + 1\right)}\right)_+.$$

For any $\epsilon' \in (0, 1)$, by choosing $\epsilon = \frac{\log(2/\epsilon')}{d-1}$, for large enough $d$, we have

$$\mathbb{E}_{z \sim \pi_0}\left[\left(\lambda - \frac{\pi_\delta}{\pi_0}(z)\right)_+ \mid \|z\|_\infty = a, \|z + \delta^*\|_\infty = a + r\right]$$

$$\geq \frac{1}{2}\left(1 - (1 - \epsilon)^{d-1}\right)\left(\lambda - \left(1 - \epsilon + \frac{r}{a}\right)^{-k} e^{-\frac{d-1-k}{2C^2}(2(1-\epsilon)a/r + 1)} e^{\frac{a^2 \log(2/\epsilon')}{C^2 r^2}}\right)_+$$

$$= \frac{1}{2}\left(1 - (1 - \frac{\log(2/\epsilon')}{d-1})^{d-1}\right)\left(\lambda - \left(1 - \frac{\log(2/\epsilon')}{d-1} + \frac{r}{a}\right)^{-k} e^{-\frac{d-1-k}{2C^2}(2(1-\epsilon)a/r + 1)} e^{\frac{a^2 \log(2/\epsilon')}{C^2 r^2}}\right)_+$$

$$\geq \frac{1}{2}(1 - \epsilon')\left(\lambda - \left(1 - \epsilon + \frac{r}{a}\right)^{-k} e^{-\frac{d-1-k}{2C^2}(2(1-\epsilon)a/r + 1)} e^{\frac{a^2 \log(2/\epsilon')}{C^2 r^2}}\right)_+.$$

Thus we have

$$\frac{1}{2}\mathbb{E}_a\mathbb{E}_{\boldsymbol{z}\sim\pi_{\mathbf{0}}}\left[\left(\lambda - \frac{\pi_{\boldsymbol{\delta}}}{\pi_{\mathbf{0}}}(\boldsymbol{z})\right)_+ \mid \|\boldsymbol{z}\|_\infty = a, \|\boldsymbol{z}+\boldsymbol{\delta}^*\|_\infty \neq a + r\right]$$
$$=\frac{1}{2}\left(1 - \epsilon'\right)\left(\lambda - \mathcal{O}(t^{-d})\right).$$

Combine the bounds, for large $d$, we have

$$\mathbb{D}_{\mathcal{F}_{[0,1]}}\left(\lambda\pi_{\mathbf{0}} \parallel \pi_{\boldsymbol{\delta}^*}\right) = \left(1 - \epsilon'\right)\left(\lambda - \mathcal{O}(t^{-d})\right).$$

$\square$

# B   More about Experiments

## B.1   Practical Algorithm

In this section, we give our algorithm for certification. Our target is to give a high probability bound for the solution of

$$\mathcal{L}_{\pi_{\mathbf{0}}}(\mathcal{F}_{[0,1]}, \mathcal{B}_{\ell_\infty,r}) = \max_{\lambda\geq 0}\left\{\lambda f_{\pi_{\mathbf{0}}}^\sharp - \mathbb{D}_{\mathcal{F}_{[0,1]}}\left(\lambda\pi_{\mathbf{0}} \parallel \pi_{\boldsymbol{\delta}}\right)\right\}$$

given some classifier $f^\sharp$. Following [9], the given classifier here has a binary output $\{0, 1\}$. Computing the above quantity requires us to evaluate both $f_{\pi_{\mathbf{0}}}^\sharp$ and $\mathbb{D}_{\mathcal{F}_{[0,1]}}\left(\lambda\pi_{\mathbf{0}} \parallel \pi_{\boldsymbol{\delta}}\right)$. A lower bound $\hat{p}_0$ of the former term is obtained through binominal test as [9] do, while the second term can be estimated with arbitrary accuracy using Monte Carlo samples. We perform grid search to optimize $\lambda$ and given $\lambda$, we draw $N$ i.i.d. samples from the proposed smoothing distribution $\pi_{\mathbf{0}}$ to estimate $\lambda f_{\pi_{\mathbf{0}}}^\sharp - \mathbb{D}_{\mathcal{F}_{[0,1]}}\left(\lambda\pi_{\mathbf{0}} \parallel \pi_{\boldsymbol{\delta}}\right)$. This can be achieved by the following importance sampling manner:

$$\lambda f_{\pi_{\mathbf{0}}}^\sharp - \mathbb{D}_{\mathcal{F}_{[0,1]}}\left(\lambda\pi_{\mathbf{0}} \parallel \pi_{\boldsymbol{\delta}}\right)$$
$$\geq \quad \lambda\hat{p}_0 - \int\left(\lambda - \frac{\pi_{\boldsymbol{\delta}}}{\pi_{\mathbf{0}}}(\boldsymbol{z})\right)_+ \pi_{\mathbf{0}}(\boldsymbol{z})d\boldsymbol{z}$$
$$\geq \quad \lambda\hat{p}_0 - \frac{1}{N}\sum_{i=1}^N\left(\lambda - \frac{\pi_{\boldsymbol{\delta}}}{\pi_{\mathbf{0}}}(\boldsymbol{z}_i)\right)_+ - \epsilon.$$

And we use reject sampling to obtain samples from $\pi_{\mathbf{0}}$. Notice that, we restrict the search space of $\lambda$ to a finite compact set so the importance samples is bounded. Since the Monte Carlo estimation is not exact with an error $\epsilon$, we give a high probability concentration lower bound of the estimator. Algorithm 1 summarized our algorithm.

---

**Algorithm 1** Certification algorithm

---

**Input:** input image $\boldsymbol{x}_0$; original classifier: $f^\sharp$; smoothing distribution $\pi_{\boldsymbol{0}}$; radius $r$; search interval
$[\lambda_{\text{start}}, \lambda_{\text{end}}]$ of $\lambda$; search precision $h$ for optimizing $\lambda$; number of samples $N_1$ for testing $p_0$;
pre-defined error threshold $\epsilon$; significant level $\alpha$;
compute search space for $\lambda$ : $\Lambda =$range$(\lambda_{\text{start}}, \lambda_{\text{end}}, h)$
compute $N_2$: number of Monte Carlo estimation given $\epsilon, \alpha$ and $\Lambda$
compute optimal disturb: $\boldsymbol{\delta}$ depends on specific setting
**for** $\lambda$ *in* $\Lambda$ **do**
$\quad$ sample $\boldsymbol{z}_1, \cdots, \boldsymbol{z}_{N_1} \sim \pi_{\boldsymbol{0}}$
$\quad$ compute $n_1 = \frac{1}{N_1} \sum_{i=1}^{N_1} f^\sharp(\boldsymbol{x}_0 + \boldsymbol{z}_i)$
$\quad$ compute $\hat{p}_0 =$LowerConfBound$(n_1, N_1, 1 - \alpha)$
$\quad$ sample $\boldsymbol{z}_1, \cdots, \boldsymbol{z}_{N_2} \sim \pi_{\boldsymbol{0}}$
$\quad$ compute $\hat{\mathbb{D}}_{\mathcal{F}_{[0,1]}} (\lambda\pi_{\boldsymbol{0}} \parallel \pi_{\boldsymbol{\delta}}) = \frac{1}{N_2} \sum_{i=1}^{N_2} \left( \lambda - \frac{\pi_{\boldsymbol{\delta}}}{\pi_{\boldsymbol{0}}}(\boldsymbol{z}_i) \right)_+$
$\quad$ compute confidence lower bound $b_\lambda = \lambda\hat{p}_0 - \hat{\mathbb{D}}_{\mathcal{F}_{[0,1]}} (\lambda\pi_{\boldsymbol{0}} \parallel \pi_{\boldsymbol{\delta}}) - \epsilon$
**end**
**if** $\max_{\lambda \in \Lambda} b_\lambda \geq 1/2$ **then**
$\quad$ $\boldsymbol{x}_0$ can be certified
**else**
$\quad$ $\boldsymbol{x}_0$ cannot be certified
**end**

---

The LowerConfBound function performs a binominal test as described in [9]. The $\epsilon$ in Algorithm 1 is given by concentration inequality.

**Theorem 7.** *Let* $h(z_1, \cdots, z_N) = \frac{1}{N} \sum_{i=1}^N \left( \lambda - \frac{\pi_{\boldsymbol{\delta}}(z_i)}{\pi_{\boldsymbol{0}}(z_i)} \right)_+$, *we yield*

$$\Pr\{|h(z_1, \cdots, z_N) - \int (\lambda\pi_{\boldsymbol{0}}(z) - \pi_{\boldsymbol{\delta}}(z))_+ \, dz| \geq \varepsilon\} \leq \exp \left( \frac{-2N\varepsilon^2}{\lambda^2} \right).$$

*Proof.* Given *McDiarmid's Inequality*, which says

$$\sup_{x_1, x_2, \ldots, x_n, \hat{x}_i} |h(x_1, x_2, \ldots, x_n) - h(x_1, x_2, \ldots, x_{i-1}, \hat{x}_i, x_{i+1}, \ldots, x_n)| \leq c_i \quad \text{for} \quad 1 \leq i \leq n,$$

we have $c_i = \frac{\lambda}{N}$, and then obtain

$$\Pr\{|h(z_1, \cdots, z_N) - \int (\lambda\pi_{\boldsymbol{0}}(z) - \pi_{\boldsymbol{\delta}}(z))_+ \, dz| \geq \varepsilon\} \leq \exp \left( \frac{-2N\varepsilon^2}{\lambda^2} \right).$$

$\square$

The above theorem tells us that, once $\epsilon, \lambda, N$ is given, we can yield a bound with high-probability $1 - \alpha$. One can also get $N$ when $\epsilon, \lambda, \alpha$ is provided. Note that this is the same as the Hoeffding bound mentioned in Section 4.2 as Micdiarmid bound is a generalization of Hoeffding bound.

However, in practice we can use a small trick as below to certify with much less comupation:

---
**Algorithm 2** Practical certification algorithm
---
**Input:** input image $x_0$; original classifier: $f^\sharp$; smoothing distribution $\pi_0$; radius $r$; search interval for $\lambda$: $[\lambda_{\text{start}}, \lambda_{\text{end}}]$; search precision $h$ for optimizing $\lambda$; number of Monte Carlo for first estimation: $N_1^0, N_2^0$; number of samples $N_1$ for a second test of $p_0$; pre-defined error threshold $\epsilon$; significant level $\alpha$; optimal perturbation $\delta$ ($\delta = [r, 0, \ldots, 0]^\top$ for $\ell_2$ attacking and $\delta = [r, \ldots, r]^\top$ for $\ell_\infty$ attacking).

**for** $\lambda$ *in* $\Lambda$ **do**

    sample $z_1, \cdots, z_{N_1^0} \sim \pi_0$

    compute $n_1^0 = \frac{1}{N_1^0} \sum_{i=1}^{N_1^0} f^\sharp(x_0 + z_i)$

    compute $\hat{p}_0 =$ LowerConfBound$(n_1^0, N_1^0, 1 - \alpha)$

    sample $z_1, \cdots, z_{N_2^0} \sim \pi_0$

    compute $\hat{\mathbb{D}}_{\mathcal{F}_{[0,1]}} (\lambda \pi_0 \| \pi_\delta) = \frac{1}{N_2^0} \sum_{i=1}^{N_2^0} \left( \lambda - \frac{\pi_\delta}{\pi_0}(z_i) \right)_+$

    compute confidence lower bound $b_\lambda = \lambda \hat{p}_0 - \hat{\mathbb{D}}_{\mathcal{F}_{[0,1]}} (\lambda \pi_0 \| \pi_\delta)$

**end**

  compute $\hat{\lambda} = \arg\max_{\lambda \in \Lambda} b_\lambda$

  compute $N_2$: number of Monte Carlo estimation given $\epsilon, \alpha$ and $\hat{\lambda}$

  sample $z_1, \cdots, z_{N_1} \sim \pi_0$

  compute $n_1 = \frac{1}{N_1} \sum_{i=1}^{N_1} f^\sharp(x_0 + z_i)$

  compute $\hat{p}_0 =$ LowerConfBound$(n_1, N_1, 1 - \alpha)$

  sample $z_1, \cdots, z_{N_2} \sim \pi_0$

  compute $\hat{\mathbb{D}}_{\mathcal{F}_{[0,1]}} (\lambda \pi_0 \| \pi_\delta) = \frac{1}{N_2} \sum_{i=1}^{N_2} \left( \lambda - \frac{\pi_\delta}{\pi_0}(z_i) \right)_+$

  compute $b = \hat{\lambda} \hat{p}_0 - \hat{\mathbb{D}}_{\mathcal{F}_{[0,1]}} (\lambda \pi_0 \| \pi_\delta) - \epsilon$

  **if** $b \geq 1/2$ **then**

    |   $x_0$ can be certified

  **else**

    |   $x_0$ cannot be certified

  **end**
---

Algorithm 2 allow one to begin with small $N_1^0, N_2^0$ to obtain the first estimation and choose a $\hat{\lambda}$. Then a rigorous lower bound can be achieved with $\hat{\lambda}$ with enough (i.e. $N_1, N_2$) Monte Carlo samples.

## B.2 Experiment Settings

The details of our method are shown in the supplementary material. Since our method requires Monte Carlo approximation, we draw $0.1M$ samples from $\pi_0$ and construct $\alpha = 99.9\%$ confidence lower bounds of that in Equation (9). The optimization on $\lambda$ is solved using grid search. For $\ell_2$ attacks, we set $k = 500$ for CIFAR-10 and $k = 50000$ for ImageNet in our non-Gaussian smoothing distribution Equation (11). If the used model was trained with a Gaussian perturbation noise of $\mathcal{N}(0, \sigma_0^2)$, then the $\sigma$ parameter of our smoothing distribution is set to be $\sqrt{(d-1)/(d-1-k)}\sigma_0$, such that the expectation of the norm $\|z\|_2$ under our non-Gaussian distribution Equation (11) matches with the norm of $\mathcal{N}(0, \sigma_0^2)$. For $\ell_1$ situation, we keep the same rule for hyperparameter selection as $\ell_2$ case, in order to make the norm of proposed distribution has the same mean with original distribution. For $\ell_\infty$ situation, we set $k = 250$ and $\sigma$ also equals to $\sqrt{(d-1)/(d-1-k)}\sigma_0$ for the mixed norm smoothing distribution Equation (13) just for consistency. More ablation study about $k$ is deferred to Appendix B.3.

## B.3 Abalation Study

On CIFAR10, we also do ablation study to show the influence of different $k$ for the $\ell_2$ certification case as shown in Table 4.

| $\ell_2$ Radius | 0.25 | 0.5 | 0.75 | 1.0 | 1.25 | 1.5 | 1.75 | 2.0 | 2.25 |
|---|---|---|---|---|---|---|---|---|---|
| Baseline (%) | 60 | 43 | 34 | 23 | 17 | 14 | 12 | 10 | 8 |
| $k = 100$ (%) | 60 | 43 | 34 | 23 | 18 | 15 | 12 | 10 | 8 |
| $k = 200$ (%) | 60 | 44 | 36 | 24 | 18 | 15 | 13 | 10 | 8 |
| $k = 500$ (%) | **61** | **46** | **37** | **25** | **19** | **16** | 14 | 11 | **9** |
| $k = 1000$ (%) | 59 | 44 | 36 | **25** | **19** | **16** | 14 | 11 | **9** |
| $k = 2000$ (%) | 56 | 41 | 35 | 24 | **19** | **16** | **15** | **12** | **9** |

Table 4: Certified top-1 accuracy of the best classifiers on cifar10 at various $\ell_2$ radius. We use the same model as [9] and do not train any new models.

## C  Illumination about Bilateral Condition[3]

The results in the main context is obtained under binary classfication setting. Here we show it has a natural generalization to multi-class classification setting. Suppose the given classifier $f^\sharp$ classifies an input $\boldsymbol{x}_0$ correctly to class A, i.e.,

$$f_A^\sharp(\boldsymbol{x}_0) > \max_{B \neq A} f_B^\sharp(\boldsymbol{x}_0) \tag{14}$$

where $f_B^\sharp(\boldsymbol{x}_0)$ denotes the prediction confidence of any class $B$ different from ground truth label $A$. Notice that $f_A^\sharp(\boldsymbol{x}_0) + \sum_{B \neq A} f_B^\sharp(\boldsymbol{x}_0) = 1$, so the necessary and sufficient condition for correct binary classification $f_A^\sharp(\boldsymbol{x}_0) > 1/2$ becomes a *sufficient* condition for multi-class prediction.

Similarly, the necessary and sufficient condition for correct classification of the *smoothed* classifier is

$$\min_{f \in \mathcal{F}} \left\{ \mathbb{E}_{\boldsymbol{z} \sim \pi_{\mathbf{0}}}[f_A(\boldsymbol{x}_0 + \boldsymbol{\delta} + \boldsymbol{z})] \quad \text{s.t.} \quad \mathbb{E}_{\pi_{\mathbf{0}}}[f_A(\boldsymbol{x}_0)] = f_{\pi_{\mathbf{0}},A}^\sharp(\boldsymbol{x}_0) \right\} >$$

$$\max_{f \in \mathcal{F}} \left\{ \mathbb{E}_{\boldsymbol{z} \sim \pi_{\mathbf{0}}}[f_B(\boldsymbol{x}_0 + \boldsymbol{\delta} + \boldsymbol{z})] \quad \text{s.t.} \quad \mathbb{E}_{\pi_{\mathbf{0}}}[f_B(\boldsymbol{x}_0)] = f_{\pi_{\mathbf{0}},B}^\sharp(\boldsymbol{x}_0) \right\}$$

for $\forall B \neq A$ and any perturbation $\boldsymbol{\delta} \in \mathcal{B}$. Writing out their Langragian forms makes things clear:

$$\max_{\lambda} \lambda f_{\pi_{\mathbf{0}},A}^\sharp(\boldsymbol{x}_0) - \mathbb{D}_{\mathcal{F}_{[0,1]}}(\lambda \pi_{\mathbf{0}} \parallel \pi_{\boldsymbol{\delta}}) > \min_{\lambda} \max_{B \neq A} \lambda f_{\pi_{\mathbf{0}},B}^\sharp(\boldsymbol{x}_0) + \mathbb{D}_{\mathcal{F}_{[0,1]}}(\pi_{\boldsymbol{\delta}} \parallel \lambda \pi_{\mathbf{0}})$$

Thus the overall necessary and sufficient condition is

$$\min_{\boldsymbol{\delta} \in \mathcal{B}} \left\{ \max_{\lambda} \left( \lambda f_{\pi_{\mathbf{0}},A}^\sharp(\boldsymbol{x}_0) - \mathbb{D}_{\mathcal{F}_{[0,1]}}(\lambda \pi_{\mathbf{0}} \parallel \pi_{\boldsymbol{\delta}}) \right) - \max_{B \neq A} \min_{\lambda} \left( \lambda f_{\pi_{\mathbf{0}},B}^\sharp(\boldsymbol{x}_0) + \mathbb{D}_{\mathcal{F}_{[0,1]}}(\pi_{\boldsymbol{\delta}} \parallel \lambda \pi_{\mathbf{0}}) \right) \right\} > 0$$

Optimizing this *bilateral* object will *theoretically give a better certification result* than our method in main context, especially when the number of classes is large. But we do not use this bilateral formulation as reasons stated below.

When both $\pi_{\mathbf{0}}$ and $\pi_{\boldsymbol{\delta}}$ are gaussian, which is [9]'s setting, this condition is equivalent to:

$$\min_{\boldsymbol{\delta} \in \mathcal{B}} \left\{ \Phi \left( \Phi^{-1}(f_{\pi_{\mathbf{0}},A}^\sharp(\boldsymbol{x}_0)) - \frac{\|\boldsymbol{\delta}\|_2}{\sigma} \right) - \max_{B \neq A} \Phi \left( \Phi^{-1}(f_{\pi_{\mathbf{0}},B}^\sharp(\boldsymbol{x}_0)) + \frac{\|\boldsymbol{\delta}\|_2}{\sigma} \right) \right\} > 0$$

$$\Leftrightarrow \quad \Phi^{-1}(f_{\pi_{\mathbf{0}},A}^\sharp(\boldsymbol{x}_0)) - \frac{r}{\sigma} > \Phi^{-1}(f_{\pi_{\mathbf{0}},B}^\sharp(\boldsymbol{x}_0)) + \frac{r}{\sigma}, \quad \forall B \neq A$$

$$\Leftrightarrow \quad r < \frac{\sigma}{2} \left( \Phi^{-1}(f_{\pi_{\mathbf{0}},A}^\sharp(\boldsymbol{x}_0)) - \Phi^{-1}(f_{\pi_{\mathbf{0}},B}^\sharp(\boldsymbol{x}_0)) \right), \forall B \neq A$$

with a similar derivation process like Appendix A.3. This is exactly the same bound in the (restated) theorem 1 of [9].

[9] use $1 - \underline{p_A}$ as a naive estimate of the upper bound of $f_{\pi_{\mathbf{0}},B}^\sharp(\boldsymbol{x}_0)$, where $\underline{p_A}$ is a lower bound of $f_{\pi_{\mathbf{0}},A}^\sharp(\boldsymbol{x}_0)$. This leads the confidence bound decay to the bound one can get in binary case, i.e., $r \leq \sigma \Phi^{-1}(f_{\pi_{\mathbf{0}},A}^\sharp(\boldsymbol{x}_0))$.

As the two important baselines [9, 10] do not take the bilateral form, we also do not use this form in experiments for fairness.

## Footnotes

[3]In fact, the theoretical part of [15] share some similar discussion with this section.