[Reviews · NeurIPS 2020]

Review 1

Summary and Contributions: This paper proposes a framework of random smoothing with non-Gaussian noise and general types of attacks, from the Lagrangian-dual perspective. The new families of non-Gaussian smoothing distributions are claimed to enjoy better performance than the Gaussian ones for certifying L_1, L_2 and L_infty robustness.

Strengths: 1. The paper is easy to understand. The illustration figures like Figures 1 and 2 make the region certification clearer. 2. The paper has both theoretical and practical contributions. In particular, the theoretical contributions are novel, which leads to a new method by non-Gaussian smoothing that achieves better certification results than previous works. After all, theoretically-principled methodology is rare in machine learning. 3. The problems/goals studied in this paper are important in the ML community. Random smoothing is a SOTA method in certified robustness. It is good to see one can improve Gaussian smoothing by looking at other distributions. But according to the recent lower bound in [30,31,32], I think the improvement might be only a constant factor.

Weaknesses: 1. I don't think the experiments (Table 3) in Section 5.2 are comparing with the strongest baseline for certifying l_infty robustness. For example, for large perturbation (e.g. 8/255 or 16/255), IBP based methods (see "On the Effectiveness of Interval Bound Propagation for Training Verifiably Robust Models") are known to significantly outperform random-smoothing based methods (e.g., ~32% certified accuracy on CIFAR-10 for 8/255 radius even though the paper was published 2 years ago). Even for certifying small perturbation radius like 2/255, the paper still does not compare with the strongest baseline: a better certified accuracy for 2/255 is around 63% in the same setup (e.g., [31]), while you report 60%. 2. [30,31,32] have shown that Gaussian noise is optimal to certify l_p robustness for p>2 up to constant factor, while the paper claims that they can improve the certified robustness for p>2 by looking at a better noise distribution than Gaussian. To me, the only explanation for this claim is that the improvement is tiny (i.e., they only improve a small constant factor). 3. The design in Section 3.2 (i.e., the involvement of dual form) and the design of new noise distribution in Section 4 seem to be two totally different elements to improve the robustness. It is unclear to me which element play a more important role. In other words, the ablation study about the two elements is missing.

Correctness: The theoretical and empirical analysis seem correct to me, though I do not check every detail and their code.

Clarity: The paper is well-written and easy to understand.

Relation to Prior Work: The paper fails to give sufficient credits to the prior works on the trade-off between robustness and accuracy in Section 3.3, which has been well-understood in the existing work.

Reproducibility: Yes

Additional Feedback: =========comment after rebuttal============= I have read the rebuttal, and it has alleviated my concern. I therefore leave my score positive.


Review 2

Summary and Contributions: -- Presents a Lagrange duality-based framework for deriving randomized smoothing certificates. -- Shows that this framework recovers the certificates from two prior works. -- With this framework, actually computing the bound requires solving a maximization problem over the perturbation \delta, which is intractable in general. -- Introduces two new distributions for L1 and L2 smoothing, respectively; these are generalizations of the Laplace and Gaussian distributions and come with a tunable parameter which can reduce to the Laplace / Gaussian special case. -- Proves that for these two distributions, the proposed framework is tractable in practice, since the optimization problem over \delta which we need to solve has a known solution. -- Proposes a smoothing distribution tailored to the L-infinity norm, and shows that there is a closed-form solution for "delta" in this case too. -- Demonstrates that their L1 and L2 distributions outperform the Gaussian / Laplace baselines for L1 / L2 robustness, and demonstrates that their L-infinity distribution outperforms Gaussian smoothing for L-infinity robustness. -- Analyze a simple toy scenario where their L-infinity solution provably outperforms Gaussian smoothing.

Strengths: The framework is interesting and potentially will be useful to others, and the L-infinity results are interesting too.

Weaknesses: The recent paper "Randomized smoothing of all shapes and sizes" presents an even wider variety of smoothing distributions than are proposed here.

Correctness: They seem correct but I did not check the math thoroughly.

Clarity: Yes.

Relation to Prior Work: I'd like to see (much) more discussion of the similarities between your framework and that of Dvijotham et al, "A framework for robustness certification of smoothed classifiers using F-divergences", https://openreview.net/forum?id=SJlKrkSFPH.

Reproducibility: Yes

Additional Feedback: -- For the L1 and L2 experiments where you showed that you outperformed the baseline, how did you set k? For each radius did you pick the best k? (In that case, it's not surprising that you outperformed the baseline, it's guaranteed -- but good to empirically demonstrate anyway.) -- You should mention in the main paper that you use rejection sampling to sample from the exotic smoothing distributions. I had to look in the appendix to find this. Update: I have read the rebuttal and am keeping my score.


Review 3

Summary and Contributions: => This paper tackles the issue of building certified defenses against l_1, l_2, and l_\infty adversarial examples. It focuses on the family of defenses that smooth the undefended classifier by convolution with some probability distribution (a.k.a. Randomized smoothing). => Most existing works on Randomized smoothing investigate simple distributions (such as Gaussian or Laplace). These distributions are known to have pathological behaviors in high-dimensional spaces leading to suboptimal certificates. This paper presents a general framework for analyzing Randomized smoothing for arbitrary distributions through the lens of functional optimization. This framework allows the authors to both reproduce some existing results, and present new bounds when Randomized smoothing is computed by using generalized exponential distributions. This theoretical work is validated through empirical evaluation on CIFAR10 and Imagenet.

Strengths: => The paper is well written and the claims are clearly stated. => The problem of building certified defenses to adversarial examples is a hot topic in machine learning and is highly relevant to the NeurIPS community. => Both the theoretical grounding and the empirical evaluation look sound to me. => To the best of my knowledge, the theoretical framework is new and it provides interesting perspectives on certified defenses based on Randomized smoothing. Moreover, this is very nice to see something that differs from classical proofs (e.g. using Neyman Pearson Lemma).

Weaknesses: => To my point of view, the main weakness of this paper is the significance of the results. Even though I agree that the general framework using functional optimization is new, the idea of using generalized exponential distributions is neither new nor the best method to improve the certified radius of Randomized smoothing (see [1] and related work section). => Similarly, the results presented the experimental part do not compare to the existing state-of-the-art. For example, to build a certified radius against l_1 adversarial attacks, the best know technique is to use uniform distribution (see [1]), which achieve better results in terms of certified accuracy than the ones presented in this paper. More generally, I believe the authors did not properly discussed previous works. [1] Randomized Smoothing of All Shapes and Sizes. Greg Yang, Tony Duan, J. Edward Hu, Hadi Salman, Ilya Razenshteyn, Jerry Li. International Conference on Machine Learning (ICML), 2020

Correctness: => As I already mentioned, the claims are clearly stated and both theoretical and empirical results look sound. I just have some minor remarks: - Everywhere in the paper, you write min/max for every optimization problem. To be fully precise these notations should not be used directly; one should keep notation sup/inf as long there is no evidence for the existence of a solution. - In Theorem 1 (I), why do \mathcal{F} and \mathcal{B} need to be compact? If it were to ensure the existence of the min, the author should prove it. Moreover, in the remaining of the paper, \mathcal{F}=\mathcal{F}_{[0,1]} which does not seem compact to me. - In Theorem 1 (II), the authors did not provide any proof. Even when the result comes from a simple calculation, I believe that any stated result should be given a proper proof.

Clarity: => The overall paper is clear and well written.

Relation to Prior Work: => The related work section looks incomplete to me. The authors did acknowledge most of the existing work on randomized smoothing, but they did not make clear how their contribution differs from previous works (with the exception of [2] and [3]). In particular, I would like to come back on [1] that the paper did cite but not for the appropriate contribution. => [1] also present an original framework for analyzing Randomized smoothing by using the Wulff Crystal of the norm used by the adversary. Based on their theoretical framework, the authors present new ways of calculating certified radii and benchmark various distributions according to the certificate they might provide. In particular, the authors investigate generalized exponential distributions to defend against l_1 and l _2 based adversaries. Here are some of their findings: - For l_2 adversaries, l_2 generalized exponential distributions do not provide much improvement w.r.t classical Gaussian distribution. Conversely, the present paper claims that l_2 generalized exponential distributions constantly provide better results. These different findings might be due to the optimization process or analytical framework, but it is crucial that the authors compared their findings to previous ones. - For l_1 Randomized smoothing, since the l_1 generalized exponential distribution has non-cubic level sets, it has little chance to provide optimal certificates. Moreover, as I already mentioned above, the uniform distribution (which has cubic level sets) provides much better certified radii. [2] Certified Adversarial Robustness via Randomized Smoothing. Jeremy M Cohen, Elan Rosenfeld, J. Zico Kolter. International Conference on Machine Learning (ICML), 2019 [3] l_1 Adversarial Robustness Certificates: a Randomized Smoothing Approach. Teng, J., Lee, G.-H., and Yuan, Y. Technical report 2019.

Reproducibility: Yes

Additional Feedback: => Even if the author did not clearly discuss previous contributions, I really enjoyed reading the paper, and the new framework was interesting to me. Hence I will advocate for acceptance of the paper. However the author should bring clear answers to the above concerns (especially comparison to [1]) for me not to degrade my notation. => Here are some typos I noticed: - line 184 'control the trade-off the accuracy' -> 'control the trade-off between accuracy' - line 191 repeated reference to Eq. 11 - Figure 3 'Pure l_2 (Eq 10)' -> 'Pure l_2 (Eq 11)' - line 451 in Supp 'delta_1' -> 'delta_i' - line 460 on the right-hand side, there is some referencing issue After rebuttal: => The author feedback answers my concerns on related works; hence I leave my rating unchanged. Including these discussions in the next version of the paper would clarify the contributions.


Review 4

Summary and Contributions: The paper studies certification of adversarial robustness via randomized smoothing inspired by recent work. The paper considers a slightly different perspective allowing for a generalization of previous approaches. They utilize the generalization to provide different smoothing schemes that provide some empirical improvement. Update: I have read the rebuttal and keep my score

Strengths: The paper addresses an important open problem of tight robustness certification. It builds up on excellent recent work and studies a generalization that is intuitive and presented well. Overall, the paper is very well-written and I found it very easy to follow all presented material including the theorem statements. The intuition behind the new smoothing distribution proposed is explained clearly. The paper also performs an empirical evaluation of their proposed new smoothing distributions, and find some small gains.

Weaknesses: I found it hard to understand the gains proposed by the new smoothing distributions, both theoretically and empirically. (i) Theoretically: it is not clear to me how the new smoothing distribution affects the robustness-accuracy tradeoff because we no longer have a closed form. For e.g. Theorem 2 does not study what the optimal value of \lambda is and whether that actually leads to a more accurate classifier, as the intuition would suggest. Similarly, the authors mention that for l_\infty certification, obtaining bounds via Gaussians corresponds to l_2 balls of very large radius (scales as square root dimension). However, it's unclear to me what the scaling (in terms of d) of the noise magnitude that the new proposed distribution admits. (ii) Empirically, on CIFAR10 and Imagenet (l2 norms), the gains seem to be around an absolute value of 1 percent. However, that seems to be within the precision errors of reported numbers. For e.g., 60 % certified accuracy of baseline vs.. 61% of paper's method. It's unclear whether this improvement is statistically significant. Related to the previous point, I wonder if this stems from the fact that the new distribution does not really lead to a better robustness-accuracy tradeoff? -- There seems to be some improvement in l_infinity numbers on real datasets and also the toy datasets. On real datasets the improvement is only significant at large \eps where the bounds are pretty bad. However, on the toy dataset, the gains seem more pronounced overall. Is there some intuition for when the bounds are tight and when not? (iii) The current SOTA in certification (via randomized smoothing) is the one by Salman et al., where they perform adversarial training. Is it possible to test adv training + the new distributions to see what the gains are?

Correctness: The theorem statements seem precise and correct. I haven't read the proofs in detail but the arguments presented seem reasonable to me. The empirical methodology is also correct, perhaps missing one experiment on combining adversarial training with the new distributions proposed. Also, just to double check, are the classifiers under "ours" trained with noise from the new distributions (like Cohen et al. trains on Gaussian distributions)?

Clarity: I found the paper quite well-written and easy to follow. Main arguments were presented well and sufficient background was provided.

Relation to Prior Work: Yes, it's clearly discussed. This work considers a slightly different perspective of randomized smoothing and provides a way to use different smoothing distributions than those considered in prior work. The intuition behind these new distributions and empirical evaluation is presented.

Reproducibility: Yes

Additional Feedback:

[Author Response · NeurIPS 2020]

We thank the reviewers for their insightful and constructive feedback. We start with addressing two common issues (**C#1**, **C#2**), and then the comments from the individual reviewers ( **R1**, **R2**, **R3**, **R4** ).

**C#1: About the connection with [A].** We will definitely draw more comprehensive discussions on [A] as suggested by **R3**. We believe the two works have different contributions and were developed concurrently. [A] derives the optimal shapes of level sets for $\ell_p$ attacks based on the Wulff Crystal theory, while our work, based on our functional-optimization framework and accuracy-robustness decomposition (Eq.9), proposes to use distribution that is more concentrated toward the center. Besides, we also consider a novel distribution using mixed $\ell_2$ and $\ell_\infty$ norm for $\ell_\infty$ adversary, which hasn't been studied before and improve the empirical results.

**C#2: About the significance of empirical results.** As we simply use the pre-trained model from open source of the baselines and thus our algorithm is almost deterministic (as the variance of Monte Carlo is very small given that we use 100K samples). Our result are statistically significant, for example for the $49\%$ vs. $50\%$ accuracy for ImageNet on $l_2$ adversarial, two-proportions z-test gives p value 0.001565.

**R1 On $\ell_\infty$ results.** We achieve better or comparable performance as IBP. We will add the comparison in revision. Previously, we do not list IBP results because it relies on a special model structure while our method can be applied to any model structure. [31] uses trades to train better model, while we focus on better certification accuracy with a given model. Applying our method to [31]'s checkpoint (image size = 32, $\epsilon = 0.435$, $\beta$=6, standard deviation = 0.12), we achieve 64.7% accuracy at 2/255 radius outperforming 62.6% in [31]. We will add this baseline in the next version.

**R1 On the optimality of Gaussian.** Yes, [30-32] gives the (rate) optimality of Gaussian for $\ell_p$ adversarial. But this theoretical results only analyzes the rate of the error while ignoring the constant. However, in practice, the constant can be important or even dominating (e.g. many non-convex optimization algorithms have same rate while can be quite different in practice due to the difference in constant factor).

**R1 On the connection between the dual form and proposed distribution.** The proposed distribution family is the one that gives empirical improvement. The theoretical framework serves as the motivation (decomposition of the bound in sec 3.3) for proposing the new distribution family. It is not valid to do ablation study on the theoretical framework (i.e. when we choose previous smoothing distribution, i.e., Gaussian in [C], our theoretical framework gives same result, see Corollary 1,2).

**R2** [B] and our work are two different ways for generalizing the original random smoothing bounds: [B] relaxes the constrain of smoothing samples to a ball in f-divergence while ours relaxes the classifier to be in a function space. We will draw an in-depth discussion in the revision. Regarding the choice of $k$, we refer readers to Appendix B.3 for details. We will move the reject sampling details to main text.

**R3 On the 'weakness'.** Thanks for the information on [A]. Regarding work [A], please see C#1. For the questions on $\ell_2$ adversarial, our improvement does not come from the optimization as we simply use the pretrain model from open source repository of the baselines. We suspect the empirical different comes from the different choice of hyper-parameters of the distribution. We are not sure what you mean by saying 'analytic form gives different findings'. Could you explain it with more detail so that we can address it in the next version? For the question on $\ell_1$ adversarial, yes, [A] achieves better empirical results. But our results support our theoretical findings and is of independent values to the community. We will definitely add comprehensive discussion on [A] in the next version.

**R3 On the minor remark about 'correctness'.** Thanks for pointing this out. We agree using sup/inf is more precise and will change to it. And yes, $\mathcal{F}_{[0,1]}$ isn't compact. We will fix this. We will also add proof for theorem 1 (III).

**R4 On the trade-off.** In line 143-147, we explain our main insights gained from Eq.9, from where we carefully design the proposed distribution to achieve a better trade-off (see Fig.1). The $\lambda$ comes from the dual derivation, and its optimal value only depends on the specific value of the accuracy term and the norm ball in Eq.9. Even though the optimal $\lambda$ is generally hard to derive analytically, the two components in Eq.9 clearly show the trade-off between accuracy and robustness. Similarly, performing rigorous scaling analysis on $\ell_\infty$ is rather challenging technically. We leave this analysis for future work.

**R4 On empirical result.** Please see C#2 on the statistical significance. The tightness of our bound is proved in theorem 1 (III). Empirically, the exact certified accuracy relies on $f_{\pi_0}^\#$, which depends on whether a good smoothing distribution that fits the data and base classifier are selected (which is also the main gain of this paper).

**R4 On adversarial training.** Thanks for pointing out this. We agree it's doable while it would require an unaffordable amount of computational resource for adv. training on ImageNet. Thus we leave this for future work.

**Reference** [A] Randomized Smoothing of All Shapes and Sizes. [B] A Framework for Robustness Certification of Smoothed Classifiers Using f-Divergences. [C] Certified Adversarial Robustness via Randomized Smoothing.

[Meta-Review · NeurIPS 2020]

Reviewers found both theoretical and practical contributions of the paper significant. However, Authors should make sure to address the comments raised by the reviewers in the final draft.